# Dopaminergic Degeneration Differentially Modulates Primary Motor Cortex Activity and Motor Behavior in Hemiparkinsonian Rats

**DOI:** 10.3390/brainsci15101123

**Published:** 2025-10-18

**Authors:** Suelen L. Boschen, Julian Seethaler, Shaohua Wang, Wendy D. Lujan, Jodi L. Silvernail, Launia J. White, Michael G. Heckman, Rickey E. Carter, Su-Youne Chang, J. Luis Lujan

**Affiliations:** 1Department of Neurologic Surgery, Mayo Clinic, Rochester, MN 55901, USA; julian.seethaler@gmail.com (J.S.); shaohua.wang3@nih.gov (S.W.); lujan.wendy@mayo.edu (W.D.L.); silvernail.jodi2@mayo.edu (J.L.S.); chang.suyoune@mayo.edu (S.-Y.C.); lujan.luis@mayo.edu (J.L.L.); 2Department of Neuroscience, Mayo Clinic, Jacksonville, FL 32224, USA; 3Neurobiology Lab, National Institute of Environmental Health Sciences, RTP, Durham, NC 27709, USA; 4Department of Qualitative Health Sciences, Mayo Clinic, Jacksonville, FL 32224, USA; white.launia@mayo.edu (L.J.W.); heckman.michael@mayo.edu (M.G.H.); carter.rickey@mayo.edu (R.E.C.); 5Department of Biomedical Engineering and Physiology, Mayo Clinic, Rochester, MN 55901, USA

**Keywords:** Parkinson’s disease (PD), primary motor cortex (M1), in vivo calcium imaging, neuronal calcium activity, dopaminergic transmission

## Abstract

Background/Goal: Parkinson’s disease (PD) disrupts dopaminergic transmission, leading to motor deficits and altered activity in the primary motor cortex (M1). While M1 modulation is critical for motor control, its response to dopaminergic degeneration and treatment remains unclear. This study aimed to characterize M1 neuronal activity and motor behavior in hemiparkinsonian rats using in vivo calcium imaging across naïve, lesioned, and levodopa-treated states. Methods: Thirteen Sprague Dawley rats were injected with GCaMP6f in the M1 and implanted with a GRIN lens and guide cannula targeting the medial forebrain bundle. Calcium imaging and motor behavior were assessed longitudinally using a single pellet reaching test (SPRT) before and after unilateral 6-hydroxydopamine (6-OHDA) lesioning and subsequent levodopa/carbidopa treatment. Dopaminergic lesion severity was quantified via tyrosine hydroxylase immunohistochemistry. Calcium event frequency and influx were analyzed with CNMF-E and statistical modeling. Results: Levodopa treatment improved fine motor performance as shown by a significant reduction in grasp errors (mean difference: −8.91, 95% CI: −16.66 to −1.16, *p* = 0.031) and increased reaching duration (mean difference: 4.13, 95% CI: 0.94 to 7.32, *p* = 0.019) compared to the lesioned state. M1 calcium activity showed modulation dependent on lesion severity: low-lesion rats exhibited reduced event frequency (mean difference: 0.04 Hz, 95% CI: 0.001 to 0.08, *p* = 0.045) and increased influx post-lesion (mean difference: −0.20 z·s, 95% CI: −0.38 to −0.02, *p* = 0.038), while high-lesion rats showed increased influx only after levodopa treatment (mean difference: −0.34 z·s, 95% CI: −0.52 to −0.16, *p* = 0.003). Correlation analyses revealed that calcium influx, but not frequency, was negatively correlated with lesion severity during levodopa treatment (Spearman r = −0.857, *p* = 0.024). Conclusion: M1 neuronal activity appears to be differentially modulated by dopaminergic degeneration and levodopa treatment in a lesion-dependent manner. These preliminary findings suggest dynamic cortical responses in PD and support the utility of calcium imaging for monitoring circuit-level changes in disease and therapy. Further research with larger cohorts and complementary methodologies will be necessary to validate and extend these observations.

## 1. Introduction

Parkinson’s disease (PD) is a progressive neurodegenerative disease that primarily affects the nigrostriatal dopaminergic pathway, resulting in movement abnormalities, including bradykinesia, tremor, and muscular rigidity [1,2,3]. The pathophysiology of PD spreads from the nigrostriatal pathway through the cortical–basal ganglia–thalamic motor network. This results in complex changes in activity in several regions, including the primary motor cortex (M1) [4,5,6].

The M1 is critically important in motor learning and volitional movement control as it is responsible for initiating downstream motor activation [6,7,8]. Dopaminergic inputs from the ventral tegmental area (VTA) and, to a lesser extent, from the substantia nigra pars compacta (SNc) directly modulate M1 neuronal activity via D1-like (D1R) and D2-like (D2R) dopaminergic receptors [9,10], affecting the firing rate and synchronization of M1 neurons. Indirect modulation of M1 activity is also achieved via SNc dopaminergic projections onto the direct and indirect pathways of the basal ganglia [11,12,13]. In PD, the loss of SNc dopaminergic input exacerbates the indirect pathway activity while weakening the direct pathway, leading to increased inhibition of motor thalamic nuclei and decreased M1 excitation, which contributes to dysfunctional motor output [14,15,16]. However, functional imaging studies show that M1 activity may be either increased or decreased in both PD patients and animal models of PD [17,18,19,20,21,22,23,24,25]. Similarly, conflicting results from single-unit electrophysiological recordings in a non-human primate M1 show either no changes or a reduction in M1 activity after MPTP treatment [26,27,28]. In contrast, studies in hemiparkinsonian rats have shown increased synchronicity and beta-frequency oscillations between the M1 and striatum [29]. Therefore, a cohesive interpretation of the pathological changes in M1 activity in the context of pathological behaviors and therapeutic responses remains elusive.

The objective of this study is to characterize how midbrain dopaminergic degeneration can induce changes in M1 activity that can be visualized and quantified using in vivo calcium imaging in awake rats. To achieve this goal, we used single-photon fluorescent calcium imaging, a technique that combines the benefits of imaging and electrophysiological recording techniques to provide an assessment of both the systems and cellular responses in the disease and treatment states in a single analysis platform. We imaged calcium activity in the M1 of GCaMP6f-expressing rats as they transitioned from a naïve baseline to a 6-hydroxydopamine (6-OHDA)-lesioned hemiparkinsonian state and then to a levodopa-treated state over the course of three months. Motor function and M1 calcium activity were chronically evaluated in the single pellet reaching test (SPRT) (Figure 1a). Here, we demonstrate that levodopa treatment of 6-OHDA-lesioned rats not only improves fine motor abilities but modulates M1 neuronal calcium activity in a manner dependent on the levels of dopaminergic lesion.

## 2. Materials and Methods

### 2.1. Animals

We used a total of 13 adult (8–9 weeks old at the beginning of the experimental design) Sprague Dawley rats, including 8 males and 5 females, with an approximate weight of 250–280 g. The rats were kept on a standard 12 h light/dark cycle in single housing at a constant 21 °C temperature and 45% humidity with ad libitum access to water and food. After approval by the Mayo Clinic Institutional Animal Care and Use Committee (IACUC), all animal procedures and experiments were conducted following the terms and guidelines of the National Institutes of Health for the use of animals and complied with the ARRIVE guidelines.

All animals underwent all three treatment states (naïve, 6-OHDA-lesioned, and levodopa-treated) in a fixed sequential order. Therefore, randomization into separate treatment groups was not applicable, as each animal served as its own control across conditions. To minimize bias and ensure reproducibility, investigators responsible for behavioral scoring and data analysis were blinded to the treatment state of the animals. This blinding was maintained throughout the quantification of motor performance and calcium imaging data.

### 2.2. General Surgical Procedures

Anesthesia was induced with 5% isoflurane and maintained with 1–2% isoflurane via a nose cone. Anesthetized rats were placed on a heated stage (37 °C), and the skull was secured with blunt ear bars and a nose clamp incisor bar in a stereotaxic frame (Kopf Systems, Tujunga, CA, USA). Analgesia was provided pre-operatively with a 1.0 mg/kg buprenorphine extended-release injection.

#### 2.2.1. Viral Transduction Procedure

On day 1, the rat’s head was shaved and disinfected with betadine, and a midline scalp 1.5–2 cm incision starting between the eyes and extending caudally to the level of the ears was performed. The periosteum was removed, and the skull was cleaned with 0.9% saline and sterile cotton swabs. A 0.5 mm burr hole was made above the M1 (AP +2.5 mm and ML +2.6 mm from bregma, DV −1.8 mm from dura mater). Coordinates for all brain regions targeted in the surgical procedures were determined using a rat brain atlas [30]. The dura mater was incised to allow insertion of a 28-gauge needle and infusion of 500 nL of 110 vg/mL genetically encoded calcium indicator pENN.AAV9-CaMKII-GcaMP6f-WPRE-SV40 (AddGene Inc, Watertown, MA, USA) at a rate of 100 nL/min. The needle was maintained in place for 5 min to prevent reflux and then withdrawn at a rate of 10 µm/s. The burr hole was covered with bone wax (Medtronic, Minneapolis, MN, USA), and the skin incision was closed in one layer with simple interrupted stitches using 4-0 vicryl sutures. Rats were continuously monitored in a heated cage and returned to their home cages once they became ambulatory.

#### 2.2.2. Guide Cannula and GRIN Lens Implantation Procedure

On day 14, rats were implanted with a guide cannula over the MFB, and a GRIN prism lens (Inscopix, Palo Alto, CA, USA) was implanted in the M1 (AP +2.5 mm and ML +2.4 mm from bregma—200 μm lateral to previous injection site to respect the lens’s working distance; DV −2.0 mm from dura mater to position center of lens around the center of the injection site). Anesthesia, analgesia, and surgical approach were identical to those described for the viral transfection procedure. The burr hole above the M1 was widened to approximately 1.5 mm. A 1 mm dorsoventral incision was made in the cortex with a straight-edge #11 dissection knife attached to a stereotactic arm perpendicular to the brain surface, allowing insertion of a 1 mm GRIN prism lens into the M1. The exposed areas surrounding the lens were covered with a biocompatible silicone elastomer Kwik-Sil (World Precision Instruments, Sarasota, FL, USA). A second burr hole (1.5 mm) was drilled to insert a 22-gauge, 10 mm guide cannula 2.0 mm above the MFB (AP −4.4 mm and ML +1.6 mm from bregma, DV −7.4 mm from dura mater). Two skull screws were placed bilaterally anterior to the bregma, and two additional screws were placed posterior to the bregma on the contralateral side of the lens and guide cannula to secure a headcap built with Metabond quick adhesive (Parkell, Edgewood, NY, USA).

#### 2.2.3. Baseplating Procedure

On day 21, a baseplate (~0.5 g) was attached above the GRIN lens to allow for attachment and support of a miniature microscope. Rats were anesthetized and secured on the stereotaxic frame as described previously, but no analgesia was provided since this is a non-invasive procedure. Isoflurane concentration was modulated to allow for visualization of calcium activity from M1 neurons via the GRIN lens to ensure correct placement of the baseplate as the baseplate was moved along three degrees of freedom (ML, AP, DV) in order to identify the location with maximal fluorescence. Once this location was identified, the baseplate was secured in place using Metabond.

### 2.3. Food Restriction

Rats were put on a food-restricted diet of 10–15 g standard food chow in order to entice them to grasp and eat sucrose pellets during the SPRT. Food restriction started 3 days prior to the first training session and was maintained during testing for a maximum of 3 weeks or until rats reached 90% of their initial body weight. Rats were weighed daily during the food restriction period.

### 2.4. Chamber for Assessment of Motor Function

Motor function was assessed via SPRT using a custom-built 360 × 80 mm chamber with transparent plastic sidewalls and a metal grid fence on one end, across which a plastic platform (88 × 45 × 25 cm) with receptacles for sucrose pellets was located just outside the fence. The fence openings were just wide enough for the rat to thrust out its forelimbs and grasp a single 45 mg sucrose pellet at a time.

### 2.5. SPRT Assessments of Motor Function

#### 2.5.1. Habituation and Training Sessions

Training sessions started 5–7 days following baseplate implantation. One day before training, rats were habituated to the testing chamber for 5 min while receiving sucrose pellets. During the training sessions, rats were gently guided to reach through the fence and grasp sucrose pellets placed on the platform receptacle. The SPRT consisted of 25 trials in which the rat was allowed to grasp and eat a total of 25 sucrose pellets; trials continued until the rats ate 25 pellets or the time limit of 15 min expired. Each trial started with the placement of a sucrose pellet in the platform receptacle and ended after the rat ate the pellet. If the rat pushed the pellet beyond its reach, a new sucrose pellet was placed in the platform and was considered the same trial. Rats were connected to a dummy miniature microscope during the training sessions to habituate the animals to the weight and size of the microscope used during testing.

#### 2.5.2. Testing Sessions

Test sessions began once rats were able to successfully grasp and eat 25 pellets within 15 min (usually after 3 to 5 days of training). Testing was conducted for 3 to 5 consecutive days, where rats were connected to a skull-mounted miniature microscope to record calcium activity from the M1. Synchronized calcium activity and pellet reach behavioral data were acquired during test sessions.

The number of attempts in a given trial to successfully grasp and eat a sucrose pellet was not limited (Figure 1b). An investigator blinded to the treatment analyzed and quantified the rats’ forelimb movements according to the following criteria:•Full grasp: Extension of paw beyond the fence and grasping of a pellet. The success ratio was calculated based on categorization of full grasps as successful if the rat ate the pellet or failure if the rat did not eat or dropped the pellet after grasping it.•Reach without grasp: Extension of the paw beyond the fence allowing it to touch the pellet but not grasping it.•Grasp without pellet: Extension of paw beyond the fence to attempt to grasp a pellet in the absence of a sucrose pellet in the receptacle.•Number of attempts/trial (Performance): The total amount of full grasps, reaches without grasps, and grasps without pellets executed by the rat in each trial. The number of attempts per trial is an indication of the rat’s performance, i.e., the lower the number of attempts per trial, the better the performance.•Attempt duration: Starts when paw lifts from the floor, extends beyond the fence, and ends when pellet is brought to mouth (full grasp) or when paw is placed back on the floor (reach without grasp).•Reaching duration: Starts when paw lifts from the floor, extends beyond the fence, and ends when the pellet is grasped.•Grasping duration: Starts when pellet is grasped and ends when pellet is put in mouth or dropped.

### 2.6. 6-OHDA Lesion and Levodopa Treatment

Rats were unilaterally infused through the previously implanted guide cannula with 3 µL of 6-OHDA hydrobromide (4 µg/µL solution of 0.02% sterile ascorbate saline) at a rate of 0.5 µL/min with 5 min for diffusion into the MFB the day after SPRT test sessions in the naïve state. To prevent 6-OHDA uptake by noradrenergic neurons, desipramine hydrochloride (25 mg/kg i.p.) was administered 30 min before 6-OHDA infusion. A period of 3 weeks was allowed following 6-OHDA lesioning for dopaminergic lesion stabilization. After stabilization and another sequence of SPRT and calcium imaging recording sessions, rats received a daily 10 mg/mL levodopa (Sigma Millipore, Burlington, MA, USA, CAG1361009) intraperitoneal (i.p.) injection co-administered with a 1.25 mg/mL carbidopa decarboxylase inhibitor (Sigma Millipore, Burlington, MA, USA, CAG1095506) dissolved in 0.9% saline for 14 days to restore dopaminergic levels while preventing peripheral levodopa degradation and increasing central concentration. The doses of levodopa/carbidopa and the duration of treatment were chosen based on the previous studies demonstrating restoration of motor function in rats while avoiding levodopa-induced dyskinesia (LID) [31,32,33,34,35]. SPRT habituation and training were re-initiated after 3–5 days of levodopa/carbidopa treatment.

### 2.7. Histology

At the end of all experiments, rats received an i.p. overdose of pentobarbital sodium (100 mg/kg) and were transcardially perfused with 0.1 M phosphate-buffered saline (PBS) followed by 4% paraformaldehyde. Rats were decapitated, their scalps exposed, skulls opened, and their brains extracted and fixed overnight in 4% paraformaldehyde. Brains were stored in 30% glycerol. Cryosectioning with 40 µm slices was performed on a sliding microtome (Leica Biosystems, Wetzlar, Germany). Tissue sections were stored in 0.1% Sodium Azide in 0.1 M PBS.

Coronal M1 slices were mounted onto glass slides and coverslipped with VectaShield containing DAPI (Vector Laboratories, Newark, CA, USA) to assess PRISM lens placement and GCaMP6f expression. If the location of the PRISM lens or GCaMP6f expression was not in the M1 region, the corresponding rats were excluded from the analysis. To quantify dopaminergic cell loss, we performed immunohistochemical staining using tyrosine hydroxylase (TH). Coronal slices of the SNc were rinsed in 0.01 M PBS with 0.2% Triton X-100 (PBS-Tx), incubated in 3% H_2_O_2_ for 10 min to quench endogenous peroxidase activity, and blocked in 10% normal goat serum (NGS) in PBS-Tx for 1 h. Slices were incubated with the primary antibody, anti-TH (rabbit polyclonal, Abcam, ab112, 1:4000), overnight at 4 °C on a shaker. Following primary antibody incubation, sections were rinsed in PBS-Tx and incubated in 10% NGS with the biotinylated goat anti-rabbit IgG secondary antibody (Vector Laboratories, B10001.5, 1:500) for 1 h at room temperature. Following the rinse in PBS-Tx, slices were incubated with an avidin–biotin enzyme complex (VECTASTAIN^®^ Elite^®^ ABC-HRP kit, PK-6101, Vector Laboratories, Newark, CA, USA), rinsed in 0.1 M PBS, and incubated with 3,3’-Diaminobenzidine chromogen (DAB Substrate kit, SK-4100, Vector Laboratories). After a final rinse in 0.1 M PBS, slices were mounted onto glass slides and coverslipped with mounting media (Eukitt^®^ Quick-hardening mounting medium, 03989, Sigma-Aldrich, St. Loius, MO, USA). Slides were imaged under a microscope (Keyence, BZ-X800, Itasca, IL, USA).

### 2.8. Characterization of Dopaminergic Lesion

Dopaminergic neuron loss was assessed by bilateral quantification of TH-positive neurons in the SNc, as previously shown [36,37,38,39]. Two to four SNc-containing midbrain sections from each animal were evaluated, and the relative ratio of TH+ cells was determined as the percentage of TH+ cells in the lesioned SNc (right side) relative to the intact SNc (left side). Midbrain sections of R13 were damaged, and densitometry analysis of TH+ terminals from four striatal sections was conducted to evaluate the level of dopaminergic depletion for the animal (Appendix A).

### 2.9. Calcium Data Collection

Cortical calcium imaging was performed using an nVista miniature microscope (Inscopix, Palo Alto, CA, USA) weighing approximately 2 g and attached to the baseplate on the rat’s head while rats moved freely in the SPRT chamber. Real-time imaging was transmitted to a computer via a data acquisition (DAQ) system connected to the miniature microscope. Each test session rendered one or more videos of calcium activity. Thus, each state (naïve, 6-OHDA-lesioned, and levodopa-treated) produced at least four videos for data analysis.

### 2.10. Calcium Data Analysis

Calcium data for each state were compared, and only the videos with the same field of view were stitched to allow uniform preprocessing and longitudinal registration of the neuroanatomy. Calcium activity data across states were not processed longitudinally due to differences in field of view. Each calcium dataset was preprocessed by spatially downsampling each video (by a factor of two), cropping the field of view to the region with increased calcium signals, and motion correcting using the Turboreg algorithm implemented in the Inscopix Data Processing Software 1.9.4 (IDPS-MOSAIC, Inscopix, Palo Alto, CA, USA, RRID:SCR_017408). Data were then exported as ISXD files and loaded into MATLAB, v. 2023 (The MathWorks, Natick, MA, USA, RRID:SCR_001622) to extract putative neurons or regions of interest (ROIs) using a constrained non-negative matrix factorization (CNMF-E) algorithm [40]. The ring model of background fluorescence was used for all datasets. Acceptance parameters were selected based on the data and included neuron diameter ≥ 7 px and signal-to-noise ratio (SNR) between 3 and 5. Next, the CNMF-E-detected ROIs were loaded into IDPS for visual inspection of the shape and traces of each ROI, and additional inclusion criteria were applied by the observer to ensure that (i) a single ROI outlined only one neuron, (ii) a single neuron was not labeled by multiple ROIs, and (iii) selected ROIs were present in all videos within each state. ROIs that did not meet these criteria were excluded from further analysis.

A peak detection algorithm was used to extract individual calcium events from raw calcium intensity traces for each ROI in MATLAB. Additionally, the area under the curve (AUC) was calculated to measure the total calcium influx using the event onset and offset of each event, indicating the activity level for each specific neuron. Calcium intensity for selected ROIs was converted to a z-score scale to allow comparisons between rats across states. The calcium event rate was calculated using Equation (1). The average calcium influx was determined by Equation (2) in each state. Finally, event rate (Hz) and average calcium influx data were used for statistical analysis and effect comparisons between states.
(1)$$\frac{\text{total number of calcium events}}{\text{total number of frames}}\times \text{frame rate}(20\;\mathrm{Hz})$$
(2)$$\frac{\sum AUC}{\text{total number of calcium events}}$$

### 2.11. Statistical Analysis

Statistical analyses were performed using R Statistical Software (version 4.2.2; R Foundation for Statistical Computing, Vienna, Austria) and GraphPad Prism (version 9.5.1; www.graphpad.com, RRID:SCR_002798) for data visualization.

Prior to hypothesis testing, all datasets were assessed for normality using the Shapiro–Wilk test. Based on the results, appropriate non-parametric tests were selected. For behavioral and calcium imaging data, mixed-effects linear regression models were used to account for repeated measures, with state (naïve, 6-OHDA-lesioned, levodopa-treated) as a fixed effect and rat as a random effect. This approach was chosen to model within-subject variability and accommodate missing data due to technical exclusions.

To assess the relationship between lesion severity and M1 calcium activity, Spearman’s rank correlation was used to evaluate associations between the percentage of TH+ neurons and both calcium event frequency and calcium influx magnitude. Spearman’s test was selected due to the non-parametric nature of the data and the ordinal classification of lesion severity.

All statistical tests were two-sided, and *p*-values < 0.05 were considered statistically significant. Non-significant trends were defined as *p*-values between 0.05 and 0.10 and are reported as preliminary observations.

## 3. Results

### 3.1. Confirmation of GCaMP6f Expression and PRISM Lens Placement

Calcium activity data from 7 of 13 rats were included in the analysis following verification of both PRISM lens placement in the M1 and GCaMP6f expression (Figure 2a). Of the six rats excluded, one rat (R5) was removed from the analysis because the baseplate detached from the headcap, which precluded microscope connection. An additional five rats (R9–R13) were excluded because calcium signals were not detected during one or more imaging sessions. Histological analysis of GCaMP6f expression and PRISM lens localization showed that four of these rats had the PRISM lens misplaced relative to the GCaMP6f-expressing area, and one rat had an extensive inflammatory reaction with cortical damage, which hindered proper calcium imaging (see Table 1 for details).

### 3.2. Characterization of Dopaminergic Lesion

Comparison of TH+ neurons in the SNc of the 6-OHDA-lesioned hemisphere (right side) to TH+ neurons on the non-lesioned control hemisphere (left side) allowed classification of rats as (i) high lesion level if TH+ neuron loss was above 70%, (ii) mild lesion level if TH+ neuron loss was between 30% and 69%, and (iii) low lesion level if TH+ neuron loss was 0 to 29%. Four rats presented a high lesion level (R2 = 13%, R8 = 27%, R9 = 15%, R13 = 18%), five rats presented a mild lesion level (R3 = 43%, R6 = 38%, R7 = 41%, R10 = 42%, R11 = 63%), and three rats presented a low lesion level (R1 = 96%, R4 = 78%, R12 = 81%) (Figure 2b, Appendix A). Individual absolute values of TH+ neurons quantified in the intact and lesioned SNc of each rat are shown in Appendix A. The different lesion levels observed were not dependent on the sex of the rats (*p* = 0.927) (Appendix A).

### 3.3. Levodopa Improves Fine Motor Abilities in Hemiparkinsonian Rats

Following SPRT training, a set of 3–5 test sessions with 25 trials each was performed. For each session, M1 activity was recorded while rats extended either their left or right forelimb through the grid fence and grasped a sucrose pellet placed in the receptacle located just outside the behavioral setup. Their motor performance was recorded on video and synchronized to the calcium recordings. Categorization and quantification of fine motor movements was conducted by a trained investigator who was blinded to the treatment (Figure 1b, Table 2).

Levodopa treatment non-significantly improved overall performance in the SPRT compared to the 6-OHDA-lesioned state (*p* = 0.063) by slightly reducing the number of attempts executed to successfully reach and grasp the pellet in each trial (Figure 3a). The trend toward improved performance by levodopa treatment was influenced by a significant reduction in grasp movements when a sucrose pellet was not in the receptacle relative to the 6-OHDA-lesioned state (*p* = 0.030) and a non-significant decrease relative to the naïve state (*p* = 0.086) (Figure 3d). In turn, the number of full grasps was also reduced in the levodopa treatment state relative to the 6-OHDA-lesioned state (*p* = 0.031) (Figure 3b), indicating improved accuracy in the execution of voluntary movements of the limbs and paws. Incomplete movements toward the sucrose pellet (reach without grasp) were not significantly affected by levodopa treatment (Figure 3c).

The levodopa treatment state had an increased total attempt duration compared to the 6-OHDA-lesioned state (*p* = 0.030), which showed a significant decrease in attempt duration relative to the naïve state (*p* = 0.036) (Figure 3e). We divided the total attempt duration into reaching duration (i.e., time to reach and grasp the sucrose pellet) and grasping duration (i.e., time during which grasp is maintained to bring the pellet to the animal’s mouth). Levodopa treatment increased the reaching duration relative to the naïve and 6-OHDA-lesioned states (*p* = 0.024 and *p* = 0.019, respectively) (Figure 3f). However, levodopa treatment did not affect grasping duration (Figure 3g). This suggests improved motor coordination and movement control of forelimbs and paws during the levodopa treatment state. Indeed, during the 6-OHDA-lesioned state, rats exhibited frequent shorter and incomplete ballistic paw/forelimb movements that did not fit the criteria for a full extension to reach or grasp a pellet beyond the grid fence.

Preliminary sex-based analysis of behavioral outcomes revealed that male rats (*n* = 4) presented a non-significant increase in the number of attempts per trial (*p* = 0.062) in the 6-OHDA-lesioned state and a significant decrease in attempts per trial during levodopa treatment (*p* = 0.007). Such performance improvement in male rats occurred due to a significant decrease in the number of grasps when a pellet was not present in the receptacle (*p* = 0.023) and a non-significant increase in the duration to complete an attempt (*p* = 0.078). Female rats (*n* = 3) did not show significant effects in the SPRT (Figure 4, Table 3).

Regrouping the rats according to the lesion levels, our results show that full grasps significantly decreased with levodopa treatment relative to the 6-OHDA-lesioned state of rats with mild lesions (*p* = 0.037), and duration to reach the sucrose pellets non-significantly increased in rats with high (*p* = 0.092) and mild (*p* = 0.081) lesion levels relative to the 6-OHDA-lesioned state. Additionally, 6-OHDA lesions significantly decreased reaching duration in rats with a high lesion level (*p* = 0.012) (Figure 5, Table 4). Importantly, the high variability and the low number of rats in each category may have influenced these observations; therefore, results should be interpreted as preliminary.

The individual success rate, defined as the number of successful attempts in which the rat grasped and ate a sucrose pellet divided by the total number of attempts, was qualitatively evaluated to determine the effect of dopaminergic lesions on motor behavior. Rats with a high lesion level (R8, R9, R13) showed a decreased success rate after the 6-OHDA lesion, followed by an increased success rate after levodopa treatment. This pattern was not observed in rats with mild lesions (R6, R7, R10, R11) (Figure 6).

Rats 1 through 5 received two days of training during the naïve state and a total of 15 trials per session (Table 1). These rats did not receive sufficient SPRT training during the naïve state, which could bias comparisons in subsequent states [7,41], and were thus analyzed separately. For these rats (R1–R5), we observed reduced performance in the SPRT following a 6-OHDA lesion, as shown by a significant increase in the number of attempts per trial (*p* = 0.040) and a non-significant trend toward an increase in full grasps (*p* = 0.097). Levodopa treatment did not change the motor outcome in the SPRT of these rats. Additionally, the success rate for rats 1–5 does not follow a similar pattern to rats 6–13. Specifically, rat 2 does not show a clear pattern of reduced success rate during the 6-OHDA-lesioned state, as expected from rats with high levels of dopaminergic neuronal loss (Appendix A).

The rotation test was conducted at the end of the SPRT in each state. Apomorphine treatment in the naïve state did not promote a preference for contralateral rotations, except for rat 8, which presented a contralateral rotation preference of 85%. In the 6-OHDA-lesioned state, apomorphine treatment increased contralateral rotations to 100% in rats with mild and high lesions but not in rats with low lesions. Levodopa treatment maintains contralateral rotation preference in rats with high lesions and results in contralateral rotation preference varying between 50% and 100% in rats with mild and low lesions (Appendix A).

### 3.4. Changes in M1 Calcium Activity in Response to Dopaminergic Lesion and Levodopa Treatment

We recorded calcium signals from M1 neurons using a miniature head-mounted microscope while rats executed the SPRT. We assessed global changes in calcium activity by determining the total number of calcium events, the frequency of calcium events (Hz), and the magnitude of calcium influx using the average of the area under the curve (AUC) of each event. We obtained consistent calcium signals recorded from 13 (minimum) to 205 (maximum) neurons across all three states for seven rats (Table 1, Appendix A). We successfully maintained focal planes and tracked neuronal ensembles for 3–4 subsequent days within each state. However, we were unable to track the same neuronal ensembles across the three states due to intrinsic technical limitations of the device, surgical approach, and natural skull growth that creates slight shifts in the field of view over a prolonged period of time (>2 weeks). Mixed-effects linear regression models (Table 5) showed that the 6-OHDA lesion was associated with a non-significant increase in the number of calcium events in the M1 (*p* = 0.068) but had no effect on the frequency (*p* = 0.992) or magnitude (*p* = 0.948) of calcium events relative to the naïve state. In contrast, levodopa treatment was not associated with changes in the total number of calcium events relative to the naïve or 6-OHDA-lesioned state (*p* = 0.185 and *p* = 0.563, respectively). Similarly, levodopa treatment was not associated with calcium frequency relative to the naïve or 6-OHDA-lesioned state (*p* = 0.222 and *p* = 0.225, respectively). However, a non-significant increased trend in calcium influx magnitude relative to the naïve state (*p* = 0.090) is observed, although no effect of levodopa treatment relative to the 6-OHDA lesion state is shown (*p* = 0.101) (Figure 7, Appendix A).

We further assessed calcium activity in rats with high (R2, R8), mild (R3, R6, R7), and low (R1, R4) lesion levels to determine if SNc dopaminergic transmission influences neuronal activity in the M1 (Figure 8, Table 6). The frequency of calcium events was significantly reduced for rats with low dopaminergic lesion levels during the 6-OHDA-lesioned state (*p* = 0.045), with a non-significant trend toward remaining reduced in the levodopa-treated state (*p* = 0.056). The magnitude of calcium influx increased in the 6-OHDA-lesioned state (*p* = 0.038) with no changes in the levodopa-treated state relative to the naïve and 6-OHDA-lesioned states (*p* = 0.164 and *p* = 0.367, respectively) (Figure 8a). No significant effects in the frequency and magnitude of calcium events were observed in rats with mild lesions during the 6-OHDA-lesioned and levodopa treatment states (Figure 8b). Levodopa treatment showed a non-significant trend toward reducing the frequency of calcium events in rats with high lesion levels compared to the 6-OHDA-lesioned state (*p* = 0.069) and significantly increased calcium influx magnitude compared to the naïve (*p* = 0.014) and 6-OHDA-lesioned (*p* = 0.003) states (Figure 8c).

Different patterns of M1 calcium activity are also observed among low-, mild-, and high-lesion-level rats during the execution of a successful or a failed full grasp (Table 7, Figure 9). Rats with a low lesion (R1 and R4) significantly reduced calcium event frequency for successful grasps in the 6-OHDA-lesioned state compared to the naïve state (*p* = 0.031) and significantly increased calcium event frequency for failed grasps in the levodopa-treated state compared to the 6-OHDA-lesioned state (*p* = 0.048). In contrast, the magnitude of calcium influx significantly increased in the 6-OHDA-lesioned state for both successful and failed grasps (*p* < 0.001, *p* < 0.001) and was reduced in the levodopa-treated state (*p* < 0.001 for successful grasps, *p* = 0.011 for failed grasps). Rats with a mild lesion (R3, R6, and R7) had a significant decrease in the frequency of calcium events for both successful and failed grasps (*p* < 0.001, *p* < 0.001) in the 6-OHDA-lesioned state, followed by an increase in calcium event frequency in the levodopa-treated state (*p* < 0.001 for successful grasps, *p* < 0.001 for failed grasps). No significant changes in calcium influx magnitude for both successful and failed grasps were observed in any of the states for rats with a mild lesion. As demonstrated in Figure 8, rats with a high lesion (R2 and R8) presented significant changes in calcium activity during the levodopa-treated state, but not in the 6-OHDA-lesioned state. Levodopa treatment resulted in a significantly lower frequency of calcium events for both successful and failed grasps (*p* = 0.003, *p* = 0.032) compared to the 6-OHDA-lesioned state. The frequency of calcium events for successful grasps in the levodopa-treated state was also lower than in the naïve state (*p* = 0.018). Similarly, a significantly lower magnitude of calcium influx for failed grasps in the levodopa-treated state was observed in comparison to the naïve state (*p* = 0.009). Interestingly, only rats with a high lesion presented a significant difference between successful and failed grasps during the naïve state (*p* = 0.034), which was not observed in the other states.

We further evaluated the influence of dopaminergic transmission on M1 neurons by correlating the frequency and magnitude of calcium events and percentage of remaining SNc dopaminergic neurons. Remarkably, in the naïve state, we show that there is a significant positive correlation between the frequency of neuronal calcium events and dopaminergic neurons in the SNc (r = 0.811, *p* = 0.035) and a non-significant negative correlation between calcium influx and the percentage of dopaminergic neurons (r = −0.750, *p* = 0.066). This effect is lost with the gradual and steady degeneration of dopaminergic neurons in SNc during the 6-OHDA-lesioned state. Neither the frequency of calcium events (r = 0.429, *p* = 0.354) nor the average of calcium influx (r = −0.179, *p* = 0.713) in the M1 was affected. However, when dopaminergic transmission was replenished with levodopa treatment, the negative correlation between calcium influx magnitude and the dopaminergic lesion level was restored (r = −0.857, *p* = 0.024), but not the frequency of calcium events (r = 0.643, *p* = 0.139) (Figure 10).

## 4. Discussion

M1 excitability is modulated by dopaminergic projections from the VTA and by the net effect of the direct and indirect pathways of the cortical–basal ganglia–thalamic circuitry [9,10]. In PD, dysfunctional dopaminergic signaling leads to an imbalance in the direct and indirect pathways, which would result in exacerbated inhibitory input to the M1, thought to be responsible for motor deficits [14,15,16]. However, multiple animal models and clinical studies report conflicting results, showing increased [5,17,18,24,25,42,43,44,45], decreased [28,46,47,48,49,50,51], or unchanged [26,27,41,45] activation of the M1. Here, we propose that such inconsistencies may result from studying M1 excitability at different levels of dopaminergic loss. An initial analysis of M1 activity shows a subtle trend toward M1 overexcitation following a 6-OHDA lesion (Figure 7). However, when stratifying by midbrain dopaminergic depletion levels, distinct patterns of M1 activation emerge. Specifically, rats with less than 70% dopaminergic depletion (low lesion level, Figure 8a) showed decreased frequency and increased magnitude of calcium events in response to a 6-OHDA lesion. In contrast, rats with more than 30% dopaminergic depletion (high lesion level, Figure 8c) showed a trend toward reduced calcium event frequency and a significant increase in calcium influx magnitude during levodopa treatment, with no significant changes during the lesioned state. Rats with mild (30–69%) dopaminergic depletion did not show changes in M1 activity (Figure 8b).

Calcium influx, as measured by our imaging approach, reflects the magnitude of neuronal activation, while frequency indicates how often these activations occur. In the context of dopaminergic lesions, previous studies have shown that severe loss of dopamine can disrupt both the excitability and synaptic integration of cortical neurons [5,17,18,24,25,28,42,43,44,45,46,47,48,49,50,51]. It is worth noting that the correlation analysis shows a significant decrease in calcium event frequency and a non-significant trend toward increased calcium influx as the dopaminergic lesion level increases. These correlations are lost during the 6-OHDA-lesioned state and partially restored with levodopa treatment (Figure 10). This suggests that M1 calcium activity gradually changes in frequency and magnitude according to the level of midbrain dopaminergic loss, and that such changes can be captured and more easily distinguished by refining movement analysis that is congruent with changes in M1 activity. This may indicate that, beyond a certain threshold of dopaminergic loss, the capacity for levodopa to enhance the magnitude of neuronal responses is limited, possibly due to irreversible changes in synaptic function or intrinsic neuronal properties. Recent studies have expanded our understanding of cortical dopamine signaling in PD, demonstrating laminar- and cell-type-specific changes in motor cortex function following dopamine depletion and highlighting compensatory cortical mechanisms that shape clinical severity [52,53,54]. Moreover, individual variability in response to dopamine replacement therapy is increasingly linked to secondary activation of cortical dopamine systems [55].

Notably, analysis of M1 calcium activity isolated to successful and failed full grasp motions reveals additional details during fine movement execution according to the dopaminergic lesion level (Figure 9). Rats with a low lesion maintained the pattern of M1 calcium activity changes shown in Figure 8a during successful and failed grasps after a 6-OHDA lesion, which is reversed by levodopa treatment. However, rats with a high lesion showed a decrease in calcium influx magnitude during failed grasps in the levodopa-treated state, in contrast to the increase in calcium influx averaged throughout the SPRT (Figure 8c). Moreover, rats with a mild lesion showed changes in M1 calcium frequency, but not calcium influx magnitude, for both successful and failed grasps, even though the average calcium event frequency and influx throughout the SPRT did not reveal significant M1 calcium activity changes (Figure 8b).

D1R and D2R may have similar effects in the M1 depending on the cortical layers and neuronal types in which they are expressed, as well as the anatomical origin of dopaminergic depletion [6,7,56]. However, their overall signaling effect through cortical–basal ganglia–thalamic circuitry may have opposing effects. In our study, the meso-cortical dopaminergic pathway and cortical–basal ganglia–thalamic circuitry were not severely damaged in rats with low dopaminergic depletion, but the latter might still have minor dysfunction given that direct 6-OHDA administration into the MFB would affect dopaminergic projections primarily in the nigrostriatal pathway [57,58]. Thus, subtle imbalances in the indirect pathway could result in increased M1 inhibition or no significant changes since dopaminergic meso-cortical projections might sustain M1 activity [59,60]. If dopaminergic transmission is supplemented with levodopa, significant changes would not be observed, likely because dopaminergic signaling is not sufficiently imbalanced by the lesion.

On the other hand, in a high dopaminergic lesion, severe damage to the cortical–basal ganglia–thalamic circuitry would result in decreased M1 activity [14,15,16,61] while severe damage to the meso-cortical pathway would result in increased M1 activity [59,60]. Thus, the net effect would be no noticeable changes in M1 activity. However, levodopa treatment would restore deficient dopaminergic transmission and allow for recovery of M1 activity. Interestingly, studies assessing M1 activity changes induced by deep brain stimulation (DBS) of the subthalamic nucleus (STN) report a decrease and regularization in M1 firing rate with stimulation [44], likely due to increased cortical GABAergic interneuron excitability [62,63,64]. The efficacy of both levodopa and STN DBS in alleviating parkinsonian symptoms in experimental animals and PD patients while modulating M1 activity reiterates the essential role of M1 in PD pathogenesis and supports the multi-circuitry regulation of M1 activity. Future experiments evaluating changes in M1 activity in response to STN DBS are necessary to further elucidate potential mechanisms of M1 activity modulation.

Notably, the pathophysiological process that culminates in neuronal degeneration and dopaminergic depletion is influenced by several biological processes. For example, chronic neuroinflammatory increases cellular vulnerability while the release of neurotrophic factors such as glial-derived neurotrophic factor (GDNF) promotes survival of neurons by modifying neuronal dysfunction, astrocytic activation, and inflammatory reactions under pathological conditions [65,66,67]. Therefore, investigating pathways beyond dopaminergic mechanisms could help explain the nuanced patterns of M1 activity observed.

Motor task performance can be directly influenced by dopaminergic transmission [68,69,70]. Thus, unilateral 6-OHDA lesioning of the nigrostriatal pathway should result in significant motor deficits. Notably, the rotation test, a standard behavioral test used to demonstrate functional dopaminergic imbalance, shows that rats with either a high or mild lesion level present a clear preference for contralateral rotations following apomorphine injection while rats with a low lesion level do not present such a preference (Appendix A). This confirms motor deficits resulting from imbalanced dopaminergic signaling, as supported by histological analysis of midbrain and striatal sections. However, the rotation test lacks sensitivity to detect subtle motor impairments associated with partial dopaminergic depletion.

To address this limitation, we employed the SPRT, which is widely used to assess fine motor control, particularly forelimb targeting, grasping, and retrieval success in rodent models of movement disorders [41,71,72,73,74]. Our results reveal that 6-OHDA lesioning leads to deficits in attempt duration, with rats exhibiting shorter durations for initiating and completing a reach attempt (Figure 3). Specifically, rats with mild lesions showed a reduction in overall attempt duration, suggesting impaired motor planning and execution. In rats with high lesion levels, reaching duration was significantly reduced, indicating difficulty in initiating and sustaining the reaching movement. Following levodopa treatment, these animals exhibited a trend toward increased reaching duration, suggesting partial restoration of motor control (Figure 5).

Moreover, levodopa treatment seems to improve success rate in rats with high dopaminergic lesions, indicating enhanced ability to grasp and retrieve the pellet. This improvement was not observed in mildly lesioned rats, consistent with the notion that motor deficits in PD become behaviorally evident when dopaminergic neuron loss exceeds 70%, disrupting the balance between direct and indirect cortical–basal ganglia–thalamic pathways [75].

In contrast to Metz et al. (2001), who reported no motor improvement with chronic levodopa treatment [71], our data show a non-significant trend toward improved performance and a significant reduction in the number of full grasps in the SPRT during the levodopa-treated state. This was accompanied by a reduced number of attempts required to successfully grasp and consume a pellet. Additionally, attempt duration increased after levodopa treatment, particularly when the pellet was present in the receptacle, suggesting improved movement precision and goal-directed behavior (Figure 3). These findings align with Hyland et al. (2019), who reported no changes in the terminal phase of reaching, likely due to convergence of movement velocities between lesioned and control animals during deceleration [41,76]. Our results suggest that levodopa-treated animals adopt a compensatory motor strategy that emphasizes goal achievement, with refined coordination of proximal and distal limb segments to accurately reach and grasp the pellet. This supports the therapeutic role of levodopa in restoring fine motor control through modulation of cortical–basal ganglia circuitry.

The finding that levodopa enhances motor output in highly lesioned rats suggests a threshold effect, wherein a critical level of dopaminergic depletion is required for exogenous dopamine therapy to exert significant behavioral benefits. This observation aligns with previous reports, which indicate that motor deficits and levodopa responsiveness become pronounced with severe neuronal loss [77,78,79]. Possible mechanisms underlying this lesion-dependent responsiveness include compensatory plasticity in less severely lesioned animals, altered dopamine receptor sensitivity, and circuit remodeling within the cortical–basal ganglia–thalamic network [80]. In mildly lesioned rats, residual endogenous dopamine and alternative neurotransmitter systems may mitigate motor impairment, reducing the observable effect of levodopa. These findings have important therapeutic implications, suggesting that the efficacy of levodopa may be constrained by the extent of dopaminergic loss and highlighting the need for personalized treatment strategies and adjunct therapies in early-stage PD.

Importantly, throughout the course of levodopa/carbidopa administration in our hemiparkinsonian rat model, we did not observe any signs of LID. Our dosing strategy was intentionally selected based on previous studies to restore motor function while minimizing the risk of LID [31,32,33,34,35], and animals were closely monitored for abnormal involuntary movements during and after treatment. The absence of dyskinetic behaviors in our cohort is consistent with reports that lower or carefully titrated doses of levodopa can avoid the development of LID in rodent models [81,82]. Nevertheless, LID remains a significant clinical concern in PD long-term management, and future studies with extended treatment duration, higher doses, or alternative protocols may be warranted to fully characterize its emergence and underlying mechanisms.

We report a preliminary analysis of sex-based effects in the motor performance of the SPRT (Figure 4), although no sex-based differences in lesion size are observed in the present study (Appendix A). Male rats presented a non-significant decrease in performance following a 6-OHDA lesion and a significant improvement in performance and grasps without pellets with levodopa treatment, while female rats did not show motor deficits following a 6-OHDA lesion or levodopa treatment. Sex differences in PD patients are well documented, with males showing greater cognitive and structural impairments than females [83,84] and a steeper slope in disease progression, possibly due to increased vulnerability to pathophysiological processes [85]. Remarkably, experimental models confirm the importance of mitochondrial and calcium homeostasis leading to higher susceptibility of male rodents to dopaminergic imbalance [86,87,88,89,90]. Further investigations in larger cohorts are necessary to confirm the sex-based differences in motor performance as a function of a dopaminergic lesion.

Several limitations should be considered when interpreting the findings of this study. First, the sample size was relatively small, particularly within subgroup analyses (e.g., “high,” “mild,” and “low” lesion groups), which may reduce statistical power and increase the risk of Type II errors. Accordingly, these subgroup analyses are framed as exploratory, and conclusions drawn from them should be interpreted with caution. Additionally, technical exclusions and methodological variability impacted the final dataset. Of the initial cohort of 13 rats, only 7 were included in the calcium imaging analysis due to issues such as misplaced lenses, neuroinflammation, or inconsistent behavioral training. These exclusions may have introduced bias and limited the generalizability of our findings. Importantly, the technical challenges reported in our study are inherent to in vivo calcium imaging and are not unique to our study [91,92,93,94,95], reflecting broader limitations of the methodology, particularly in lesioned or inflamed tissue. Our analysis was limited to population-level comparisons, as we were unable to track the same neuronal ensembles across different experimental states. This restricts our ability to assess longitudinal changes in individual neurons and may obscure finer dynamics of M1 activity. While single-neuron tracking was not feasible within the current experimental framework, our population-level analyses still reveal robust and reproducible state-dependent patterns of M1 activity that provide meaningful insights into circuit modulation. These findings are central to our study’s objectives and highlight the value of ensemble-level analysis in understanding dopaminergic regulation. Moreover, calcium imaging captures activity only in neurons that are actively firing, potentially under-representing quiescent populations that may be functionally relevant, especially in the lesioned state. These limitations could be addressed in future studies using complementary methods such as immediate–early gene expression or electrophysiology to capture a broader spectrum of neuronal activity.

While calcium imaging provides valuable insights into neuronal ensemble activity, it is important to recognize that calcium signals are indirect proxies for action potential firing. Calcium influx accompanies neuronal depolarization and action potentials, but the temporal resolution and amplitude of calcium transients do not always linearly reflect spike rates, especially in vivo and across different neuronal compartments [91,92]. The relationship between calcium event frequency/influx and motor output is therefore complex and context-dependent. Recent studies have shown that computational approaches can improve the inference of spike rates from calcium imaging data, but these methods are still evolving [93]. In our study, we interpret changes in calcium event frequency and influx as indicative of altered neuronal activation patterns in the primary motor cortex, which are associated with motor behavioral outcomes. However, we acknowledge that future studies combining calcium imaging with direct electrophysiological recordings and advanced modeling will be essential to further substantiate these correlations and clarify the mechanistic links between neuronal activity and motor output.

Lesion severity classification was based on TH loss percentages. While this provides a quantifiable measure of dopaminergic depletion, incorporating additional physiological criteria, such as neurochemical quantification of dopamine or electrophysiological measures, would strengthen the classification and provide a more comprehensive assessment of lesion severity. Additionally, we did not conduct a detailed analysis of ipsilateral versus contralateral forelimb use, as rats showed a strong preference for using the left paw following a 6-OHDA lesion. This behavioral bias limited our ability to assess lateralized motor deficits.

## 5. Conclusions

This study provides initial insights into the relationship between dopaminergic depletion, levodopa treatment, and M1 neuronal activity during fine motor behavior in a PD rat model. By integrating behavioral performance with in vivo calcium imaging, we observed lesion-severity-dependent changes in both the frequency and magnitude of M1 calcium events, with levodopa treatment partially restoring neuronal activation in severely lesioned animals. These findings suggest that the therapeutic efficacy of levodopa may be constrained by the extent of dopaminergic loss, potentially due to alterations in synaptic integration or intrinsic neuronal properties.

However, due to the limitations of the study, including small sample size, technical exclusions, and methodological variability, these results should be interpreted as preliminary and exploratory. The observed trends and correlations offer a valuable starting point for further investigation but do not yet support definitive conclusions. Future studies with larger cohorts, longitudinal tracking of neuronal ensembles, and complementary techniques such as electrophysiology or gene expression profiling will be essential to validate and expand upon these findings. Ultimately, this work lays the groundwork for deeper exploration into the circuit-level mechanisms underlying motor recovery and cortical modulation in PD.

## Figures and Tables

**Figure 1 brainsci-15-01123-f001:**
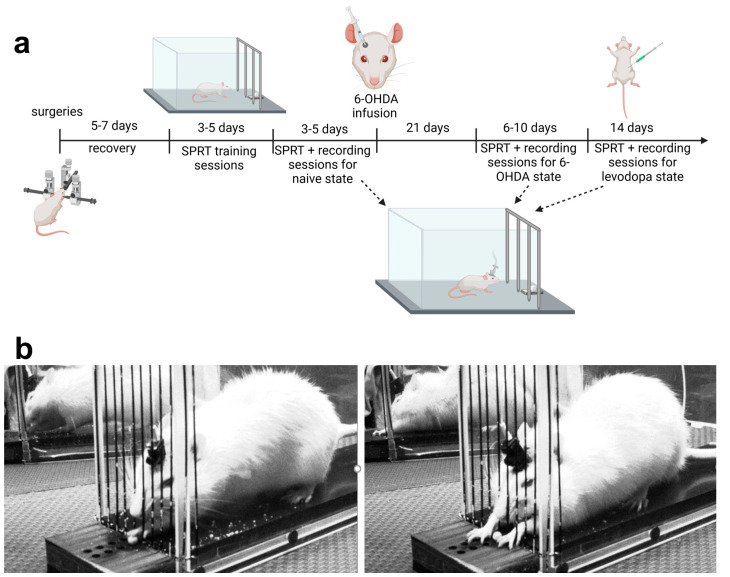
Experimental design. The experimental protocol (**a**) consisted of surgical procedures performed to inject GCaMP6f and implant a PRISM lens in the M1, implant a guide cannula in the medial forebrain bundle (MFB), and attach a baseplate for subsequent miniature microscope connection. Following recovery, longitudinal assessments of fine motor function were performed using the single pellet reaching test (SPRT) while neuronal calcium imaging was simultaneously recorded during three conditions: normal physiological state (naïve), following a unilateral intra-MFB infusion of 6-OHDA, and during the course of levodopa treatment. Once the longitudinal assessment was completed, rats were euthanized, and their brains were extracted for histological analysis. (**b**) Representative images of a full grasp (**left**) and a reach motion without grasp (**right**) executed by a hemiparkinsonian rat in the SPRT.

**Figure 2 brainsci-15-01123-f002:**
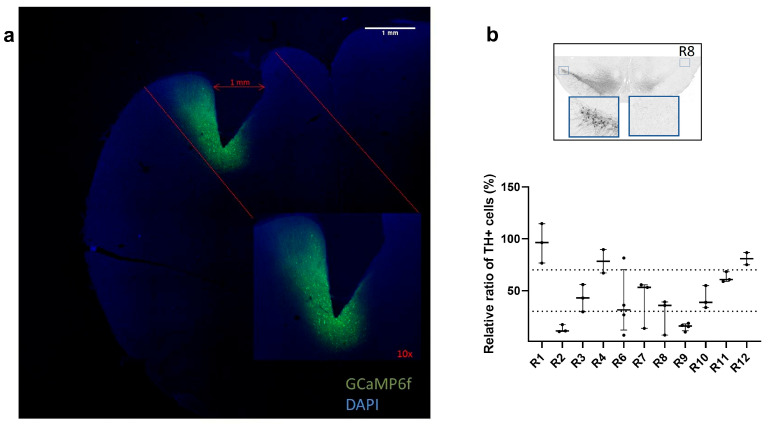
Histological assessment of fluorescent calcium indicator expression and PRISM lens implantation. (**a**) Representative image of GCaMP6f expression in M1 neurons of R3 relative to PRISM lens implantation site. (**b**) Histological quantification of TH+ neurons in the substantia nigra pars compacta (SNc) with representative midbrain image of R8. Data are shown as median ± interquartile range.

**Figure 3 brainsci-15-01123-f003:**
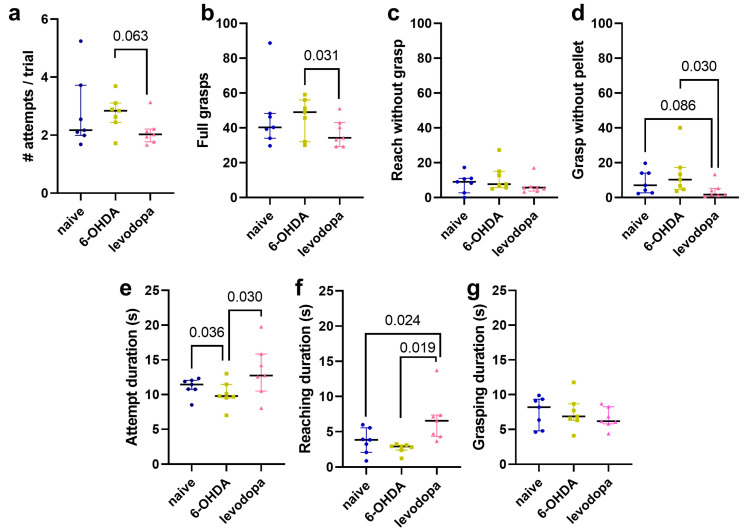
Levodopa improves fine motor abilities in hemiparkinsonian rats. Rats performed better in the single pellet reaching test (SPRT) after levodopa treatment. Overall performance measured as number of attempt in each trial (**a**). Full grasps include grasps that resulted in succesful and failed eating of a sucrose pellet (**b**). Reach without grasp are paw extensions toward the sucrose pellet but no grasping motions (**c**). Grasp without pellet include grasping motions in the absence of a pellet in the receptacle (**d**). Attempt duration is the total time to reach, grasp, and bring the pellet to mouth (**e**). Reaching duration is the time to extend the paw and reach the pellet in the receptacle (**f**) and grasping duration is the time the rat grasps the pellet and retracts the paw, bringing the pellet to its mouth (**g**). Mixed-effects linear regression models, including a fixed effect for state and a random effect for rat. Data are represented as median ± interquartile range, *n* = 7.

**Figure 4 brainsci-15-01123-f004:**
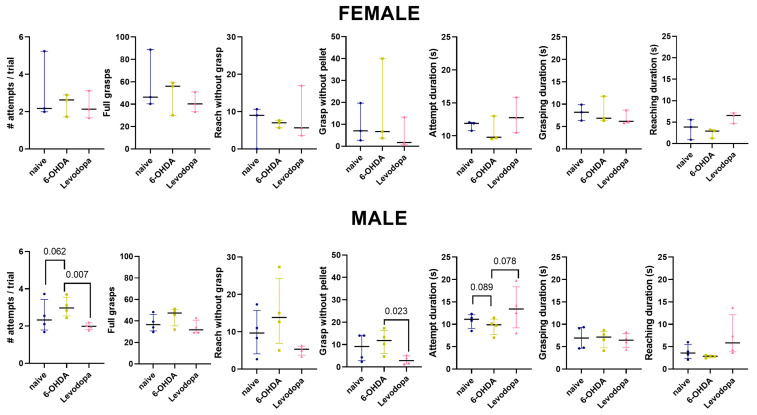
Levodopa affects fine motor abilities in male rats. Male rats presented improved performance in the SPRT during levodopa treatment. Mixed-effects linear regression models, including a fixed effect for state and a random effect for rat. Data are represented as median ± interquartile range, *n* = 4 male rats, 3 female rats.

**Figure 5 brainsci-15-01123-f005:**
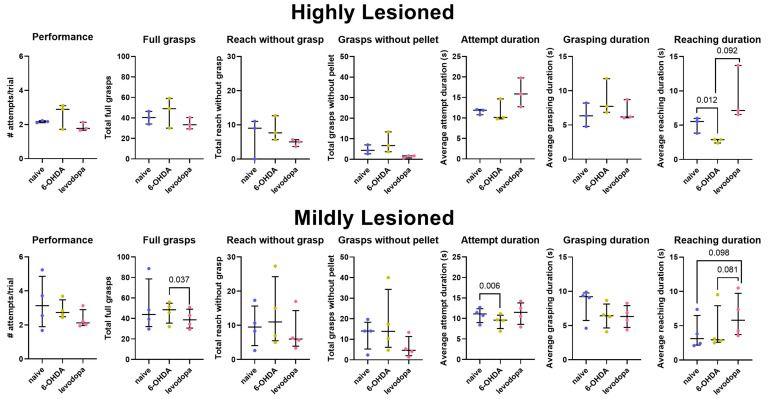
Fine motor abilities of rats with high or mild dopaminergic lesions. Levodopa treatment differentially affects fine motor abilities of rats in the SPRT after a 6-OHDA lesion. Data are represented as mean ± SD, *n* = 3 rats with a high lesion level (R8, R9, R13) and 4 rats with a mild lesion level (R6, R7, R10, R11).

**Figure 6 brainsci-15-01123-f006:**
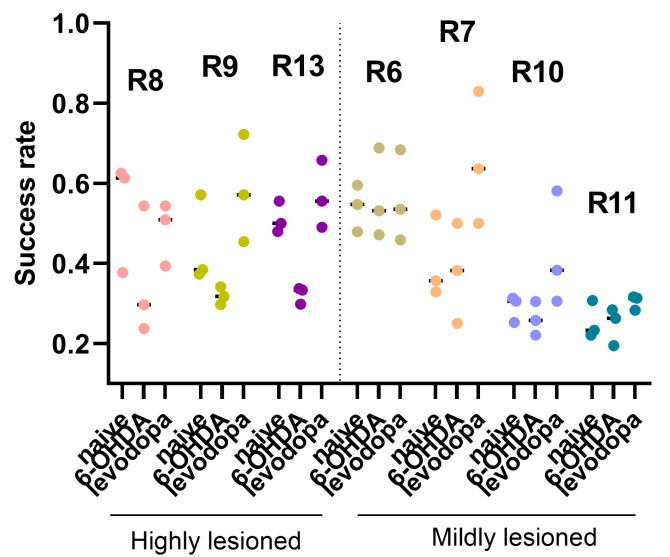
Success rate across naïve, 6-OHDA-lesioned, and levodopa-treated states. Rats with high lesion levels presented lower success rates following the 6-OHDA lesion. The success rate of these rats increased with levodopa treatment. Rats with a mild lesion did not show a similar pattern of changes in the success rate across the different states. Dot plots represent the ratio of successful attempts over total attempts per trial in each state for individual rats (n = 3 trials/state).

**Figure 7 brainsci-15-01123-f007:**
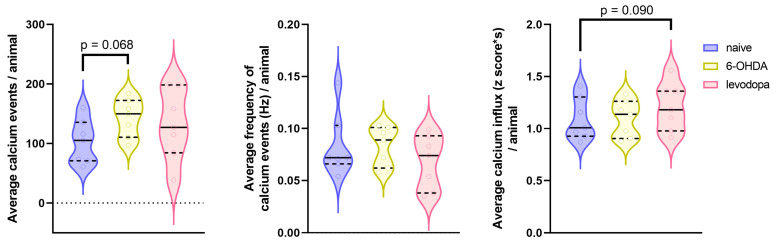
Neuronal calcium activity in the motor cortex of rats. Single-photon neuronal calcium recordings were acquired from the M1 of rats during the naïve, 6-OHDA-lesioned, and levodopa treatment states. Unilateral injection of 6-OHDA in the MFB and subsequent levodopa treatment produced changes in the M1 calcium activity. Data were analyzed using a linear mixed model with random effects for animal. The average of each rat is represented by a circle, and the median and interquartile range are represented by filled and dotted lines, respectively. The shape of the violin plots represents the distribution of individual data points. *n* = 7 rats (ranging within an average of 91 neurons recorded during the naïve state, 81 after the 6-OHDA lesion, and 92 during levodopa treatment—Appendix A).

**Figure 8 brainsci-15-01123-f008:**
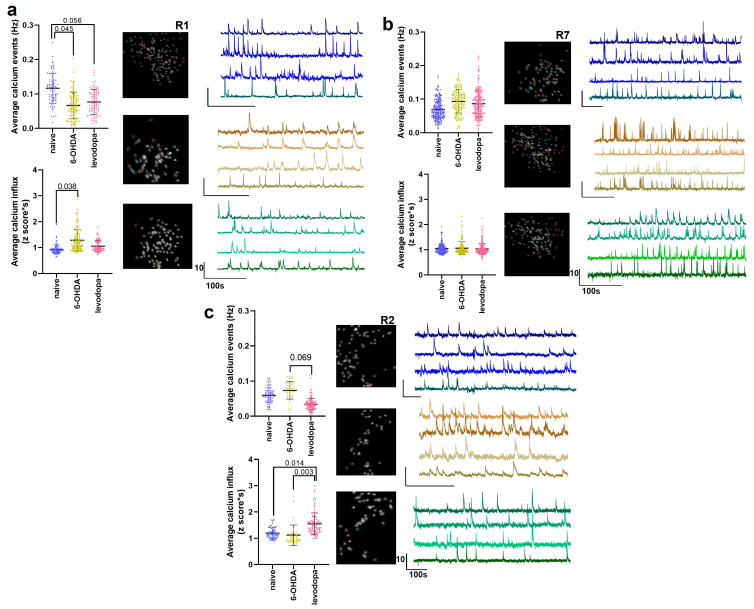
Neuronal calcium activity in the M1 of rats changes as a function of the level of dopaminergic lesions. M1 activity in rats with a high level (>70%) of dopaminergic lesions is more responsive to levodopa treatment, whereas rats with a low level of dopaminergic lesions (<30%) show changes in M1 activity after a 6-OHDA lesion that are not influenced by levodopa. Data from two rats with low lesion levels are shown in (**a**), from three rats with mild lesion levels are shown in (**b**), and from 2 rats with high lesion levels in (**c**). Data were analyzed with mixed model analysis with random factors considering the means of lesion levels and are represented as median and interquartile range. Representative images and calcium traces of rats with low lesion (R1) (**a**), mild lesion (R7) (**b**), and high lesion (R2) levels (**c**) in the naïve state (blue traces), 6-OHDA-lesioned state (brown traces), and levodopa-treated state (green traces).

**Figure 9 brainsci-15-01123-f009:**
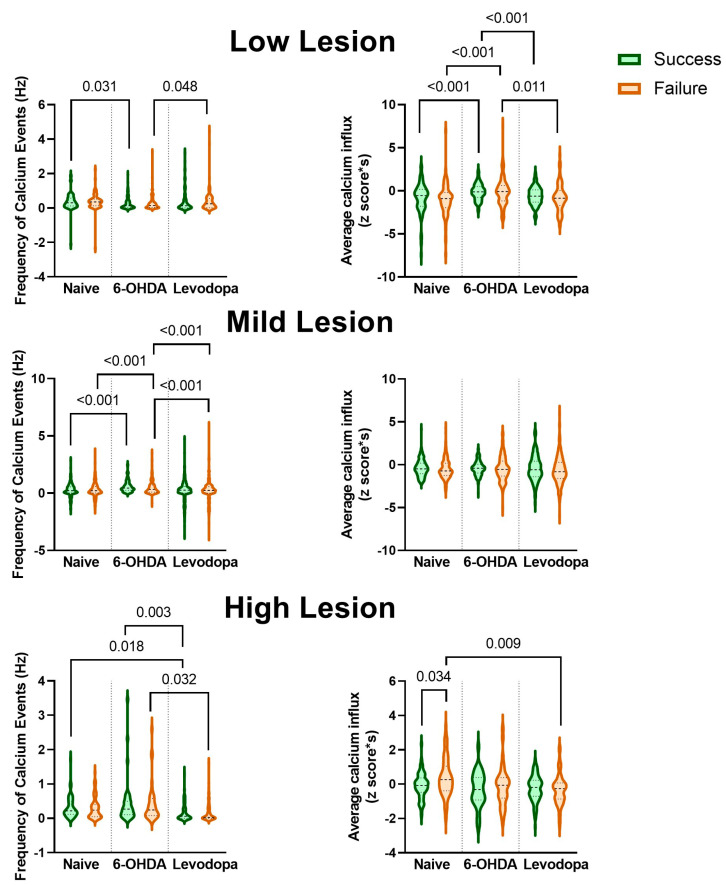
Rats with low, mild, and high levels of lesions present different patterns of neuronal calcium activity during successful and failed grasps. Calcium event frequency and magnitude of calcium influx were averaged during the execution of successful and failed full grasp motion of paws in all three states (naïve, 6-OHDA-lesioned, and levodopa-treated). Data were analyzed using a linear mixed model with a random effect for animal. The average for each rat is displayed as median and interquartile range, with the shape of the violin plots representing the distribution of individual data points.

**Figure 10 brainsci-15-01123-f010:**
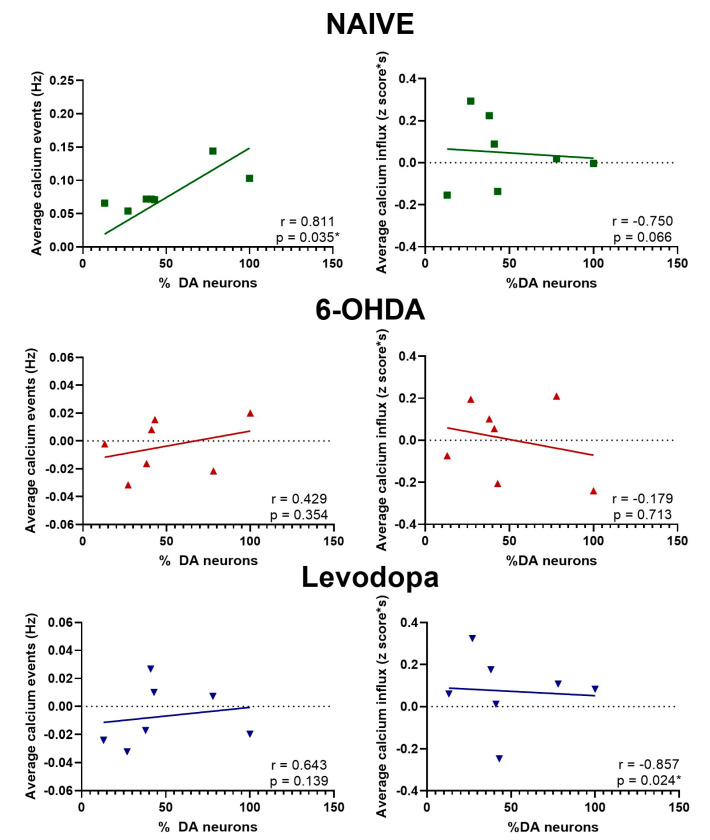
Levodopa modulates neuronal calcium activity in the M1 as a function of a dopaminergic lesion. M1 calcium event frequency and calcium influx magnitude are not significantly affected by a gradual and steady loss of dopaminergic neurons as represented in the 6-OHDA lesion state. However, upon levodopa treatment, M1 calcium influx magnitude is negatively correlated to the dopaminergic lesion level, resembling the naïve state. Spearman correlation, * *p* < 0.05.

**Table 1 brainsci-15-01123-t001:** Description of histological assessment and reasons for removal from data analysis.

Subject ID/Sex	Histology	SPRT	Ca^2+^ Activity	Reason for Removal from SPRT or Ca^2+^ Imaging Analysis
R1/male	Low TH+ loss	Yes: 15 trials	Yes	SPRT protocol different from final analysis
R2/male	High TH+ loss	Yes: 15 trials	Yes	SPRT protocol different from final analysis
R3/male	Mild TH+ loss	Yes: 15 trials	Yes	SPRT protocol different from final analysis
R4/male	Low TH+ loss	Yes: 15 trials	Yes	SPRT protocol different from final analysis
R5/female	Not assessed	Yes: 15 trials	No	SPRT protocol different from final analysis; Baseplate detached from headcap
R6/male	Mild TH+ loss	Yes: 25 trials	Yes	N/A
R7/male	Mild TH+ loss	Yes: 25 trials	Yes	N/A
R8/female	High TH+ loss	Yes: 25 trials	Yes	N/A
R9/female	High TH+ loss	Yes: 25 trials	No	No calcium activity during levodopa treatment state
R10/male	Mild TH+ loss	Yes: 25 trials	No	No calcium activity during 6-OHDA lesion and levodopa treatment states
R11/female	Mild TH+ loss	Yes: 25 trials	No	No calcium activity during 6-OHDA lesion and levodopa treatment states
R12/female	Low TH+ loss	No	No	No calcium activity in any state, inflammation in M1
R13/male	High TH+ loss in the STR	Yes: 25 trials	No	No calcium activity during levodopa treatment state

**Table 2 brainsci-15-01123-t002:** Comparison outcomes of rats 6 through 13 in the SPRT.

	Naïve vs. 6-OHDA		Naïve vs. Levodopa		6-OHDA vs. Levodopa	
Variable	Mean Difference (95% CI)	*p*-Value	Mean Difference (95% CI)	*p*-Value	Mean Difference (95% CI)	*p*-Value
Attempts/Trial (Performance)	−0.02 (−1.16, 1.12)	0.970	−0.65 (−1.47, 0.17)	0.100	−0.63 (−1.31, 0.05)	0.063
Full Grasps	−0.52 (−17.67, 16.62)	0.940	−9.43 (−22.49, 3.63)	0.130	−8.91 (−16.66, −1.16)	0.031
Reach without Grasp	3.05 (−1.89, 7.99)	0.180	−1.76 (−7.77, 4.25)	0.500	−4.81 (−13.31, 3.69)	0.220
Grasp without Pellet	1.94 (−3.10, 6.98)	0.370	−4.94 (−10.90, 1.01)	0.086	−6.89 (−12.79, −0.98)	0.030
Attempt Duration	−1.03 (−1.96, −0.10)	0.036	2.28 (−0.83, 5.38)	0.120	3.31 (0.45, 6.17)	0.030
Reaching Duration	−0.99 (−2.77, 0.79)	0.220	3.14 (0.58, 5.70)	0.024	4.13 (0.94, 7.32)	0.019
Grasping Duration	−0.04 (−2.50, 2.42)	0.970	−0.86 (−2.57, 0.84)	0.2600	−0.82 (−1.92, 0.27)	0.120

CI = confidence interval. Mean differences, 95% CIs, and *p*-values result from mixed-effects linear regression models, including a fixed effect for state and a random effect for rat. For naïve vs. 6-OHDA comparisons, mean differences are interpreted as the difference in the mean outcome level for 6-OHDA minus naïve. For naïve vs. levodopa comparisons, mean differences are interpreted as the difference in the mean outcome level for levodopa minus naïve. For 6-OHDA vs. levodopa comparisons, mean differences are interpreted as the difference in the mean outcome level for levodopa minus 6-OHDA.

**Table 3 brainsci-15-01123-t003:** Comparison outcomes of male and female rats 6 through 13 in the SPRT.

**Female**						
	Naïve vs. 6-OHDA		Naïve vs. Levodopa		6-OHDA vs. Levodopa	
Variable	Mean difference (95% CI)	*p*-value	Mean difference (95% CI)	*p*-value	Mean difference (95% CI)	*p*-value
Attempts/Trial (Performance)	−0.80 (−1.08, 2.68)	0.476	0.90 (−0.46, 2.27)	0.204	0.10 (−0.67, 0.88)	0.925
Full Grasps	10.11 (−31.95, 52.17)	0.777	16.89 (−14.49, 48.26)	0.325	6.78 (−15.17, 28.73)	0.666
Reach without Grasp	−0.22 (−17.07, 16.62)	0.999	−2.22 (−13.66, 9.21)	0.847	−2.00 (−11.64, 7.64)	0.828
Grasp without Pellet	−0.33 (−11.47, 10.80)	0.995	3.67 (−2.77, 10.10)	0.245	4.00 (−0.97, 8.97)	0.101
Attempt Duration	0.80 (−2.09, 3.69)	0.718	−1.47 (−4.52, 1.58)	0.398	−2.27 (−6.77, 2.23)	0.367
Reaching Duration	0.48 (−2.22, 3.18)	0.869	−2.48 (−6.98, 2.02)	0.310	−2.96 (−6.56, 0.64)	0.106
Grasping Duration	−0.16 (−3.43, 3.10)	0.989	1.24 (−2.17, 4.66)	0.574	1.41 (−0.75, 3.56)	0.209
**Male**						
	Naïve vs. 6-OHDA		Naïve vs. Levodopa		6-OHDA vs. Levodopa	
Variable	Mean difference (95% CI)	*p-*value	Mean difference (95% CI)	*p-*value	Mean difference (95% CI)	*p-*value
Attempts/Trial (Performance)	−0.51 (−1.04, 0.03)	0.062	0.52 (−0.16, 1.21)	0.143	1.03 (0.31, 1.75)	0.007
Full Grasps	−6.67 (−27.28, 13.95)	0.667	3.83 (−10.79, 18.46)	0.764	10.50 (−3.9, 24.9)	0.166
Reach without Grasp	−5.17 (−21.82, 11.49)	0.688	4.75 (−6.80, 16.30)	0.527	9.92 (−1.80, 21.64)	0.100
Grasp without Pellet	−2.75 (−10.13, 4.63)	0.588	5.58 (−1.02, 12.19)	0.100	8.33 (1.2, 15.47)	0.023
Attempt Duration	1.26 (−0.18, 2.69)	0.089	−2.88 (−7.45, 1.7)	0.249	−4.13 (−8.70, 0.44)	0.078
Reaching Duration	0.88 (−1.72, 3.48)	0.645	−3.63 (−8.02, 0.75)	0.108	−4.51 (−10.26, 1.23)	0.131
Grasping Duration	0.08 (−2.18, 2.34)	0.995	0.46 (−1.33, 2.25)	0.773	0.38 (−1.51, 2.28)	0.852

CI = confidence interval. Mean differences, 95% CIs, and *p*-values result from mixed-effects linear regression models, including a fixed effect for state and a random effect for rat. For naïve vs. 6-OHDA comparisons, mean differences are interpreted as the difference in the mean outcome level for 6-OHDA minus naïve. For naïve vs. levodopa comparisons, mean differences are interpreted as the difference in the mean outcome level for levodopa minus naïve. For 6-OHDA vs. levodopa comparisons, mean differences are interpreted as the difference in the mean outcome level for levodopa minus 6-OHDA.

**Table 4 brainsci-15-01123-t004:** Comparison outcomes of highly and mildly lesioned hemiparkinsonian rats in the SPRT.

**Highly Lesioned (R8, R9, R13)**						
	Naïve vs. 6-OHDA		Naïve vs. Levodopa		6-OHDA vs. Levodopa	
Variable	Mean difference (95% CI)	*p-*value	Mean difference (95% CI)	*p-*value	Mean difference (95% CI)	*p*-value
Attempts/Trial (Performance)	0.48 (−1.17, 2.14)	0.340	−0.23 (−0.65, 0.18)	0.140	−0.72 (−2.31, 0.88)	0.190
Full Grasps	5.78 (−33.89, 45.45)	0.590	−5.89 (−22.25, 10.47)	0.260	−11.67 (−43.96, 20.62)	0.260
Reach without Grasp	2.00 (−11.68, 15.68)	0.590	−1.89 (−14.29, 10.51)	0.580	−3.89 (−13.20, 5.42)	0.210
Grasp without Pellet	3.22 (−10.21, 16.65)	0.410	−3.33 (−8.95, 2.28)	0.130	−6.55 (−17.82, 4.71)	0.130
Attempt Duration	−0.61 (−4.41, 3.20)	0.560	4.55 (−4.35, 13.45)	0.160	5.16 (−4.50, 14.81)	0.150
Reaching Duration	−2.95 (−4.34, −1.55)	0.012	4.01 (−4.45, 12.47)	0.180	6.95 (−2.82, 16.73)	0.092
Grasping Duration	2.34 (−1.65, 6.33)	0.130	0.54 (−1.23, 2.31)	0.320	−1.80 (−4.77, 1.18)	0.120
**Mildly Lesioned (R6, R7, R10, R11)**						
	Naïve vs. 6-OHDA		Naïve vs. Levodopa		6-OHDA vs. Levodopa	
Variable	Mean difference (95% CI)	*p*-value	Mean difference (95% CI)	*p*-value	Mean difference (95% CI)	*p*-value
Attempts/Trial (Performance)	−0.40 (−2.80, 2.01)	0.640	−0.97 (−2.75, 0.81)	0.180	−0.57 (−1.81, 0.67)	0.240
Full Grasps	−5.25 (−35.90, 25.41)	0.620	−12.08 (−41.77, 17.60)	0.290	−6.84 (−12.88, −0.79)	0.037
Reach without Grasp	3.83 (−5.56, 13.23)	0.280	−1.67 (−14.77, 11.44)	0.710	−5.50 (−24.30, 13.31)	0.420
Grasp without Pellet	0.66 (−8.74, 10.07)	0.790	−6.56 (−24.04, 10.93)	0.250	−7.22 (−23.77, 9.33)	0.200
Attempt Duration	−1.34 (−1.97, −0.72)	0.006	0.58 (−1.80, 2.96)	0.500	1.92 (−0.88, 4.72)	0.120
Reaching Duration	0.48 (−1.65, 2.61)	0.530	2.49 (−0.85, 5.83)	0.098	2.01 (−0.46, 4.48)	0.081
Grasping Duration	−1.82 (−4.38, 0.73)	0.110	−1.91 (−4.68, 0.85)	0.120	−0.09 (−0.77, 0.59)	0.710

CI = confidence interval. Mean differences, 95% CIs, and *p*-values result from mixed-effects linear regression models, including a fixed effect for state and a random effect for rat. For naïve vs. 6-OHDA comparisons, mean differences are interpreted as the difference in the mean outcome level for 6-OHDA minus naïve. For naïve vs. levodopa comparisons, mean differences are interpreted as the difference in the mean outcome level for levodopa minus naïve. For 6-OHDA vs. levodopa comparisons, mean differences are interpreted as the difference in the mean outcome level for levodopa minus 6-OHDA.

**Table 5 brainsci-15-01123-t005:** Description of statistical analysis of neuronal calcium activity in the M1 of rats during the SPRT.

	Naïve vs. 6-OHDA		Naïve vs. Levodopa		6-OHDA vs. Levodopa	
Variable	Mean Difference (95% CI)	*p-*Value	Mean Difference (95% CI)	*p-*Value	Mean Difference (95% CI)	*p-*Value
Calcium events	−38.92 (−81.3 to 3.42)	0.068	−27.4 (−69.7 to 14.99)	0.185	11.6 (−30.8 to 53.91)	0.563
Frequency of calcium events	0.0001 (−0.02 to 0.02)	0.992	0.01 (−0.01 to 0.04)	0.222	0.01 (−0.01 to 0.04)	0.225
Calcium influx magnitude	−0.004 (−0.14 to 0.13)	0.948	−0.12 (−0.25 to 0.02)	0.090	−0.11 (−0.25 to 0.02)	0.101

CI = confidence interval. Mean differences, 95% CIs, and *p*-values result from mixed-effects linear regression models, including a fixed effect for state and a random effect for rat. For naïve vs. 6-OHDA comparisons, mean differences are interpreted as the difference in the mean outcome level for 6-OHDA minus naïve. For naïve vs. levodopa comparisons, mean differences are interpreted as the difference in the mean outcome level for levodopa minus naïve. For 6-OHDA vs. levodopa comparisons, mean differences are interpreted as the difference in the mean outcome level for levodopa minus 6-OHDA.

**Table 6 brainsci-15-01123-t006:** Description of statistical analysis of M1 neuronal calcium activity according to the levels of dopaminergic lesions.

**Low Lesion (R1, R4)**						
	Naïve vs. 6-OHDA		Naïve vs. Levodopa		6-OHDA vs. Levodopa	
Variable	Mean difference (95% CI)	*p*-value	Mean difference (95% CI)	*p*-value	Mean difference (95% CI)	*p*-value
Average Calcium Events (Hz)	0.04 (0.001, 0.08)	0.045	0.04 (−0.001, 0.08)	0.056	−0.002 (−0.04, 0.04)	0.897
Average Calcium Influx (z score*s)	−0.20 (−0.38, −0.02)	0.038	−0.12 (−0.31, 0.06)	0.164	−0.08 (−0.11, 0.26)	0.367
**Mild Lesion (R3, R6, R7)**						
	Naïve vs. 6−OHDA		Naïve vs. Levodopa		6−OHDA vs. Levodopa	
Variable	Mean difference (95% CI)	*p*-value	Mean difference (95% CI)	*p*-value	Mean difference (95% CI)	*p*-value
Average Calcium Events (Hz)	−0.02 (−0.05, 0.01)	0.238	−0.007 (−0.04, 0.03)	0.639	0.01 (−0.02, 0.04)	0.453
Average Calcium Influx (z score*s)	0.06 (−0.09, 0.21)	0.356	−0.02 (−0.17, 0.13)	0.749	−0.09 (−0.24, 0.06)	0.226
**High Lesion (R2, R8)**						
	Naïve vs. 6−OHDA		Naïve vs. Levodopa		6−OHDA vs. Levodopa	
Variable	Mean difference (95% CI)	*p*-value	Mean difference (95% CI)	*p*-value	Mean difference (95% CI)	*p*-value
Average Calcium Events (Hz)	−0.01 (−0.10, 0.03)	0.457	0.02 (−0.02, 0.06)	0.227	0.04 (−0.004, 0.08)	0.069
Average Calcium Influx (z score*s)	0.09 (−0.10, 0.27)	0.301	−0.25 (−0.43, −0.07)	0.014	−0.34 (−0.52, −0.16)	0.003

CI = confidence interval. Mean differences, 95% CIs, and *p*-values result from mixed-effects linear regression models, including a fixed effect for state and a random effect for rat. For naïve vs. 6-OHDA comparisons, mean differences are interpreted as the difference in the mean outcome level for 6-OHDA minus naïve. For naïve vs. levodopa comparisons, mean differences are interpreted as the difference in the mean outcome level for levodopa minus naïve. For 6-OHDA vs. levodopa comparisons, mean differences are interpreted as the difference in the mean outcome level for levodopa minus 6-OHDA.

**Table 7 brainsci-15-01123-t007:** Description of statistical analysis of M1 neuronal calcium activity for successful and failed full grasps.

**Frequency of Calcium Events (Hz)**						
**Low Lesion (R1, R4)**						
	**Success X Success**		**Failed X Failed**		**Success X Failed**	
Variable	Mean difference (95% CI)	*p*-value	Mean difference (95% CI)	*p*-value	Mean difference (95% CI)	*p*-value
Naïve vs. Naïve	-	-	-	-	0.04 (−0.11, 0.12)	>0.999
Naïve vs. 6-OHDA	0.12 (0.006, 0.23)	0.031	0.10 (−0.002, 0.21)	0.057	-	-
Naïve vs. Levodopa	0.02 (−0.13, 0.17)	0.100	−0.03 (−0.17, 0.10)	0.986	-	-
6-OHDA vs. 6-OHDA	-	-	-	-	−0.006 (−0.08, 0.07)	0.100
6-OHDA vs. Levodopa	−0.10 (−0.22, 0.02)	0.184	−0.14 (−0.27, −0.00)	0.048	-	-
Levodopa vs. Levodopa	-	-	-	-	−0.04 (−0.19, 0.11)	0.962
**Mild Lesion (R3, R6, R7)**						
	**Success X Success**		**Failed X Failed**		**Success X Failed**	
Variable	Mean difference (95% CI)	*p*-value	Mean difference (95% CI)	*p*-value	Mean difference (95% CI)	*p*-value
Naïve vs. Naïve	-	-	-	-	-0.03 (-0.15, 0.09)	0.100
Naïve vs. 6-OHDA	−0.41 (−0.56,−0.25)	<0.001	−0.30 (−0.47, −0.12)	<0.001	-	-
Naïve vs. Levodopa	−0.08 (−0.20, 0.04)	0.465	−0.07 (−0.21, 0.07)	0.762	-	-
6-OHDA vs. 6-OHDA	-	-	-	-	0.08 (−0.04, 0.20)	0.353
6-OHDA vs. Levodopa	0.33 (0.23, 0.43)	<0.001	0.23 (0.13, 0.32)	<0.001	-	-
Levodopa vs. Levodopa	-	-	-	-	−0.01 (−0.12, 0.09)	0.998
**High Lesion (R2, R8)**						
	**Success X Success**		**Failed X Failed**		**Success X Failed**	
Variable	Mean difference (95% CI)	*p*-value	Mean difference (95% CI)	*p*-value	Mean difference (95% CI)	*p*-value
Naïve vs. Naïve	-	-	-	-	−0.01 (−0.10, 0.09)	>0.999
Naïve vs. 6-OHDA	−0.13 (−0.32, 0.07)	0.415	−0.10 (−0.28, 0.08)	0.589	-	-
Naïve vs. Levodopa	0.11 (0.01, 0.21)	0.018	0.09 (−0.01, 0.20)	0.138	-	-
6-OHDA vs. 6-OHDA	-	-	-	-	0.02 (−0.18, 0.22)	0.100
6-OHDA vs. Levodopa	0.24 (0.05, 0.42)	0.003	0.20 (0.01, 0.38)	0.032	-	-
Levodopa vs. Levodopa	-	-	-	-	−0.02 (−0.09, 0.05)	0.969
**Average Calcium Influx**						
**Low Lesion (R1, R4)**						
	**Success X Success**		**Failed X Failed**		**Success X Failed**	
Variable	Mean difference (95% CI)	*p*-value	Mean difference (95% CI)	*p*-value	Mean difference (95% CI)	*p*-value
Naïve vs. Naïve	-	-	-	-	0.11 (−0.52, 0.74)	0.995
Naïve vs. 6-OHDA	−0.78 (−1.18, −0.39)	<0.001	−0.91 (−1.52, −0.30)	<0.001	-	-
Naïve vs. Levodopa	−0.24 (−0.71, 0.22)	0.660	−0.33 (−0.10, 0.33)	0.713	-	-
6-OHDA vs. 6-OHDA	-	-	-	-	−0.01 (−0.30, 0.27)	>0.999
6-OHDA vs. Levodopa	0.54 (0.24, 0.84)	<0.001	0.58 (0.09, 1.08)	0.011	-	-
Levodopa vs. Levodopa	-	-	-	-	0.03 (−0.38, 0.43)	>0.999
**Mild Lesion (R3, R6, R7)**						
	**Success X Success**		**Failed X Failed**		**Success X Failed**	
Variable	Mean difference (95% CI)	*p*-value	Mean difference (95% CI)	*p*-value	Mean difference (95% CI)	*p*-value
Naïve vs. Naïve	-	-	-	-	0.07 (−0.14, 0.27)	0.940
Naïve vs. 6-OHDA	−0.08 (−0.30, 0.15)	0.929	−0.09 (−0.38, −0.19)	0.944	-	-
Naïve vs. Levodopa	−0.07 (−0.34, 0.21)	0.984	0.12 (−0.22, 0.46)	0.922	-	-
6-OHDA vs. 6-OHDA	-	-	-	-	0.05 (−0.23, 0.33)	0.996
6-OHDA vs. Levodopa	0.01 (−0.31, 0.33)	>0.999	0.21 (−0.17, 0.58)	0.608	-	-
Levodopa vs. Levodopa	-	-	-	-	0.25 (−0.09, 0.59)	0.284
**High Lesion (R2, R8)**						
	**Success X Success**		**Failed X Failed**		**Success X Failed**	
Variable	Mean difference (95% CI)	*p*-value	Mean difference (95% CI)	*p*-value	Mean difference (95% CI)	*p*-value
Naïve vs. Naïve	-	-	-	-	−0.43 (−0.84, -0.02)	0.034
Naïve vs. 6-OHDA	0.22 (−0.20, 0.64)	0.649	0.44 (−0.18, 1.06)	0.329	-	-
Naïve vs. Levodopa	0.15 (−0.10, 0.40)	0.556	0.60 (0.10, 1.10)	0.009	-	-
6-OHDA vs. 6-OHDA	-	-	-	-	−0.21 (−0.70, 0.29)	0.826
6-OHDA vs. Levodopa	−0.08 (−0.42, 0.27)	0.988	0.16 (−0.40, 0.72)	0.968	-	-
Levodopa vs. Levodopa	-	-	-	-	0.02 (−0.31, 0.35)	>0.999

CI = confidence interval. Mean differences, 95% CIs, and *p*-values result from mixed-effects linear regression models, including a fixed effect for state and a random effect for rat. For naïve vs. 6-OHDA comparisons, mean differences are interpreted as the difference in the mean outcome level for 6-OHDA minus naïve. For naïve vs. levodopa comparisons, mean differences are interpreted as the difference in the mean outcome level for levodopa minus naïve. For 6-OHDA vs. levodopa comparisons, mean differences are interpreted as the difference in the mean outcome level for levodopa minus 6-OHDA.

## Data Availability

The raw data supporting the conclusions of this article will be made available by the authors on request.

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
