# Peer review of "Dopaminergic Degeneration Differentially Modulates Primary Motor Cortex Activity and Motor Behavior in Hemiparkinsonian Rats"

_brainsci, 2025, doi:10.3390/brainsci15101123_

Round 1

Reviewer 1 Report

Comments and Suggestions for Authors

Boschen et al present the significance of midbrain dopaminergic degeneration as a possible modulator of primary motor cortex activity and motor behavior. I have the following comments:

  • midbrain degeneration could be additionally discussed in the context of possible pathophysiological parameters crucial in certain parkinsonisms as Progressive Supranuclear Palsy as recently it was discussed in the context of links between neurotrophic factors and midbrain degeneration - Ref.  The significance of glial cell line-derived neurotrophic factor analysis in Progressive Supranuclear Palsy. Sci Rep. 2024;14(1):2805. Published 2024 Feb 2. doi:10.1038/s41598-024-53355-y /// Persephin-overexpressing neural stem cells regulate the function of nigral dopaminergic neurons and prevent their degeneration in a model of Parkinson's disease. Mol Cell Neurosci. 2002;21(2):205-222. doi:10.1006/mcne.2002.1171 
  • the description of improvement of fine motor performance following levodopa treatment should be more detailed
  • Authors state: "Correlation analyses revealed that calcium influx, but not frequency, was negatively associated with lesion severity during levodopa treatment." - the background of this issue should be more emphasized
  • the limitations of the study should be extensively discussed
  • due to the study's limitations authors could provide the outcome of the study as an initial point in further discussions rather than conclusive 

Author Response

Boschen et al present the significance of midbrain dopaminergic degeneration as a possible modulator of primary motor cortex activity and motor behavior. I have the following comments:

We thank the reviewer for their time to review our manuscript and use their expertise to provide meaningful feedback on our manuscript.

  • midbrain degeneration could be additionally discussed in the context of possible pathophysiological parameters crucial in certain parkinsonisms as Progressive Supranuclear Palsy as recently it was discussed in the context of links between neurotrophic factors and midbrain degeneration - Ref.  The significance of glial cell line-derived neurotrophic factor analysis in Progressive Supranuclear Palsy. Sci Rep. 2024;14(1):2805. Published 2024 Feb 2. doi:1038/s41598-024-53355-y /// Persephin-overexpressing neural stem cells regulate the function of nigral dopaminergic neurons and prevent their degeneration in a model of Parkinson's disease. Mol Cell Neurosci. 2002;21(2):205-222. doi:10.1006/mcne.2002.1171 

We thank the reviewer for this suggestion. We agree that the role of midbrain degeneration in parkinsonian syndromes warrants further discussion, particularly in the context of neurotrophic factors. In line with the reviewer’s recommendation, we have expanded our discussion in the revised manuscript (see Discussion section, page 25), to address the significance of glial cell line-derived neurotrophic factor (GDNF) in midbrain degeneration, emphasizing how neurotrophic factors may modulate neuronal vulnerability. We believe this addition strengthens the translational relevance of our work and provides a more comprehensive context for interpreting our findings.

“Notably, the pathophysiological process that culminates in neuronal degeneration and dopaminergic depletion is influenced by several biological processes. For example, chronic neuroinflammatory increases cellular vulnerability while the release of neurotrophic factors such as glial-derived neurotrophic factor (GDNF) promotes survival of neurons by modifying neuronal dysfunction, astrocytic activation and inflammatory reactions under pathological conditions.[67-69] Therefore, investigating pathways beyond dopaminergic mechanisms could help explain the nuanced patterns of M1 activity observed.”

  • the description of improvement of fine motor performance following levodopa treatment should be more detailed

We have revised the paragraphs discussing SPRT results to provide more details to explain motor performance improvement upon levodopa treatment, particularly in highly-lesioned state. Additionally, we have included a paragraph to substantiate the sex-based motor performance differences observed.

“Motor task performance can be directly influenced by dopaminergic transmission.[70-72] Thus, unilateral 6-OHDA lesioning of the nigrostriatal pathway should result in significant motor deficits. Notably, the rotation test, a standard behavioral test to demonstrate functional dopaminergic imbalance, shows that rats with either high or mild lesion level present a clear preference for contralateral rotations following apomorphine injection while rats with low lesion level did not present such preference (Supplementary Figure S4). This confirms motor deficits resulting from imbalanced dopaminergic signaling, as supported by histological analysis of midbrain and striatal sections. However, the rotation test lacks sensitivity to detect subtle motor impairments associated with partial dopaminergic depletion.

To address this limitation, we employed the SPRT, which is widely used to assess fine motor control, particularly forelimb targeting, grasping, and retrieval success in rodent models of movement disorders.[42, 73-76] Our results reveal that 6-OHDA lesioning leads to deficits in attempt duration, with rats exhibiting shorter durations to initiate and complete a reach attempt (Figure 3). Specifically, rats with mild lesions showed a reduction in overall attempt duration, suggesting impaired motor planning and execution. In rats with high lesion levels, reaching duration was significantly reduced, indicating difficulty in initiating and sustaining the reaching movement. Following levodopa treatment, these animals exhibited a trend toward increased reaching duration, suggesting partial restoration of motor control (Figure 5).

Moreover, levodopa treatment seems to improve success rate in rats with high dopaminergic lesions, indicating enhanced ability to grasp and retrieve the pellet. This improvement was not observed in mildly lesioned rats, consistent with the notion that motor deficits in PD become behaviorally evident when dopaminergic neuron loss exceeds 70%, disrupting the balance between direct and indirect cortico-basal ganglia-thalamic pathways.[77]

In contrast to Metz et al. (2001), who reported no motor improvement with chronic levodopa treatment,[78] our data show a non-significant trend toward improved performance and a significant reduction in the number of full grasps in the SPRT during the levodopa-treated state. This was accompanied by a reduced number of attempts required to successfully grasp and consume a pellet. Additionally, attempt duration increased after levodopa treatment, particularly when the pellet was present in the receptacle, suggesting improved movement precision and goal-directed behavior (Figure 3). These findings align with Hyland et al. (2019), who reported no changes in the terminal phase of reaching, likely due to convergence of movement velocities between lesioned and control animals during deceleration.[42, 79] Our results suggest that levodopa-treated animals adopt a compensatory motor strategy that emphasizes goal achievement, with refined coordination of proximal and distal limb segments to accurately reach and grasp the pellet. This supports the therapeutic role of levodopa in restoring fine motor control through modulation of cortico-basal ganglia circuitry.

The finding that levodopa enhances motor output in highly lesioned rats suggests a threshold effect, wherein a critical level of dopaminergic depletion is required for exogenous dopamine therapy to exert significant behavioral benefits. This observation aligns with previous reports, which indicate that motor deficits and levodopa responsiveness become pronounced with severe neuronal loss.[80-82] Possible mechanisms underlying this lesion-dependent responsiveness include compensatory plasticity in less severely lesioned animals, altered dopamine receptor sensitivity, and circuit remodeling within the cortico-basal ganglia-thalamic network.[83] In mildly lesioned rats, residual endogenous dopamine and alternative neurotransmitter systems may mitigate motor impairment, reducing the observable effect of levodopa. These findings have important therapeutic implications, suggesting that the efficacy of levodopa may be constrained by the extent of dopaminergic loss and highlighting the need for personalized treatment strategies and adjunct therapies in early-stage PD.

Importantly, throughout the course of levodopa/carbidopa administration in our hemi-parkinsonian rat model, we did not observe any signs of LID. Our dosing strategy was intentionally selected based on previous studies to restore motor function while minimizing the risk of LID,[32-36] and animals were closely monitored for abnormal involuntary movements during and after treatment. The absence of dyskinetic behaviors in our cohort is consistent with reports that lower or carefully titrated doses of levodopa can avoid the development of LID in rodent models.[84, 85] Nevertheless, LID remains a significant clinical concern in long-term management of Parkinson’s disease, and future studies with extended treatment duration, higher doses, or alternative protocols may be warranted to fully characterize its emergence and underlying mechanisms.

We report sex-based effects in the motor performance of the SPRT (Figure 4), although no sex-based differences in lesion size are observed in the present study (Supplementary Figure S2). Male rats presented a non-significant decrease in performance following 6-OHDA lesion and a significant improvement in performance and grasps without pellets with levodopa treatment, while female rats did not show motor deficits following 6-OHDA lesion or levodopa treatment. Sex differences in PD patients are well documented, with males showing greater cognitive and structural impairments than females,[86, 87] and steeper slope in disease progression possibly due to increased vulnerability to pathophysiological processes.[88] Remarkably, experimental models confirm the importance of mitochondrial and calcium homeostasis leading to higher susceptibility of male rodents to dopaminergic imbalance.[89-93]”

  • Authors state: "Correlation analyses revealed that calcium influx, but not frequency, was negatively associated with lesion severity during levodopa treatment." - the background of this issue should be more emphasized

We appreciate the reviewer’s request for additional background on this point. Calcium influx, as measured by our imaging approach, reflects the magnitude of neuronal activation, while frequency indicates how often these activations occur. In the context of dopaminergic lesions, previous studies have shown that severe loss of dopamine can disrupt both the excitability and synaptic integration of cortical neurons. Our finding that only calcium influx (and not frequency) is negatively associated with lesion severity during levodopa treatment suggests that the restorative effects of levodopa on neuronal activation are more pronounced in less severely lesioned animals. This may indicate that, beyond a certain threshold of dopaminergic loss, the capacity for levodopa to enhance the magnitude of neuronal responses is limited, possibly due to irreversible changes in synaptic function or intrinsic neuronal properties. We have now included this information to the discussion (see Discussion section, page 24).

  • the limitations of the study should be extensively discussed

We thank the reviewer for highlighting the importance of a thorough discussion of the study’s limitations. We have now expanded the limitations section in the revised manuscript providing a transparent and balanced assessment of the study’s limitations while guiding future research in this area.

“Several limitations should be considered when interpreting the findings of this study. First, the sample size was relatively small, particularly within subgroup analyses (e.g., “high,” “mild,” and “low” lesion groups), which may reduce statistical power and increase the risk of Type II errors. Accordingly, these subgroup analyses are framed as exploratory, and conclusions drawn from them should be interpreted with caution. Additionally, technical exclusions and methodological variability impacted the final dataset. Of the initial cohort of 13 rats, only 7 were included in the calcium imaging analysis due to issues such as misplaced lenses, neuroinflammation, or inconsistent behavioral training. These exclusions may have introduced bias and limited the generalizability of our findings. Importantly, the technical challenges reported in our study are inherent to in vivo calcium imaging and are not unique to our study,[94-98] reflecting broader limitations of the methodology, particularly in lesioned or inflamed tissue. Our analysis was limited to population-level comparisons, as we were unable to track the same neuronal ensembles across different experimental states. This restricts our ability to assess longitudinal changes in individual neurons and may obscure finer dynamics of M1 activity. While single-neuron tracking was not feasible within the current experimental framework, our population-level analyses still reveal robust and reproducible state-dependent patterns of M1 activity that provide meaningful insights into circuit modulation. These findings are central to our study’s objectives and highlight the value of ensemble-level analysis in understanding dopaminergic regulation. Moreover, calcium imaging captures activity only in neurons that are actively firing, potentially under-representing quiescent populations that may be functionally relevant, especially in the lesioned state. These limitations could be addressed in future studies using complementary methods such as immediate-early gene expression or electrophysiology to capture a broader spectrum of neuronal activity.

While calcium imaging provides valuable insights into neuronal ensemble activity, it is important to recognize that calcium signals are indirect proxies for action potential firing. Calcium influx accompanies neuronal depolarization and action potentials, but the temporal resolution and amplitude of calcium transients do not always linearly reflect spike rates, especially in vivo and across different neuronal compartments [94, 95]. The relationship between calcium event frequency/influx and motor output is therefore complex and context-dependent. Recent studies have shown that computational approaches can improve the inference of spike rates from calcium imaging data, but these methods are still evolving [96]. In our study, we interpret changes in calcium event frequency and influx as indicative of altered neuronal activation patterns in the primary motor cortex, which are associated with motor behavioral outcomes. However, we acknowledge that future studies combining calcium imaging with direct electrophysiological recordings and advanced modeling will be essential to further substantiate these correlations and clarify the mechanistic links between neuronal activity and motor output.

Lesion severity classification was based on TH loss percentages. While this provides a quantifiable measure of dopaminergic depletion, incorporating additional physiological or behavioral criteria would strengthen the classification and provide a more comprehensive assessment of lesion severity. Additionally, we did not conduct a detailed analysis of ipsilateral versus contralateral forelimb use, as rats showed a strong preference for using the left paw following 6-OHDA lesion. This behavioral bias limited our ability to assess lateralized motor deficits.”

  • due to the study's limitations authors could provide the outcome of the study as an initial point in further discussions rather than conclusive 

We thank the reviewer for this recommendation. We agree with this perspective and have tempered the discussion of our results throughout the manuscript and have stated a new conclusion that better matches the tone of the revised manuscript.

“This study provides initial insights into the relationship between dopaminergic depletion, levodopa treatment, and M1 neuronal activity during fine motor behavior in a PD rat model. By integrating behavioral performance with in vivo calcium imaging, we observed lesion severity-dependent changes in both the frequency and magnitude of M1 calcium events, with levodopa treatment partially restoring neuronal activation in severely lesioned animals. These findings suggest that the therapeutic efficacy of levodopa may be constrained by the extent of dopaminergic loss, potentially due to alterations in synaptic integration or intrinsic neuronal properties.

However, due to the limitations of the study, including small sample size, technical exclusions, and methodological variability, these results should be interpreted as preliminary and exploratory. The observed trends and correlations offer a valuable starting point for further investigation but do not yet support definitive conclusions. Future studies with larger cohorts, longitudinal tracking of neuronal ensembles, and complementary techniques such as electrophysiology or gene expression profiling will be essential to validate and expand upon these findings. Ultimately, this work lays the groundwork for deeper exploration into the circuit-level mechanisms underlying motor recovery and cortical modulation in PD.”

Reviewer 2 Report

Comments and Suggestions for Authors

This manuscript presents an interesting study in which the authors use imaging techniques to characterize M1 neuronal activity and motor behavior in a rat model of hemiparkinsonism.

The work is well written, clear and methodologically reliable. The only significant limitation is the relatively small number of animals included in the study, which may reduce the statistical power of the findings. However, the authors explicitly acknowledge this in the discussion and conclusions, demonstrating their awareness of the study's limitations.

Author Response

This manuscript presents an interesting study in which the authors use imaging techniques to characterize M1 neuronal activity and motor behavior in a rat model of hemiparkinsonism.

The work is well written, clear and methodologically reliable. The only significant limitation is the relatively small number of animals included in the study, which may reduce the statistical power of the findings. However, the authors explicitly acknowledge this in the discussion and conclusions, demonstrating their awareness of the study's limitations.

We thank the reviewer for their time in reviewing our study and providing feedback.

Reviewer 3 Report

Comments and Suggestions for Authors

This is a valuable and well-conceived study with significant translational implications. The integration of in vivo calcium imaging with behavioral analysis offers meaningful insights into how dopaminergic degeneration differentially affects primary motor cortex activity and motor function, as well as how these changes are modulated by levodopa treatment. However, the manuscript would benefit greatly from addressing the following points.

  • The authors acknowledge the small sample size and the risk of Type II errors, particularly in subgroup analyses. This limitation is critical, especially since the “high,” “mild,” and “low” lesion groups include only 2-4 animals each. Such analyses should be clearly framed as exploratory rather than confirmatory, and the conclusions drawn from them should be tempered accordingly.
  • The initial cohort consisted of 13 rats, yet only 7 were included in the calcium imaging analysis due to technical exclusions and inconsistencies in behavioral training (15 vs. 25 trials in the naïve state). These methodological differences could influence both behavioral and imaging outcomes and should be clearly justified. The authors should discuss how such variability may have affected data interpretation and overall conclusions.
  • The manuscript lacks sufficient detail on experimental design elements such as randomization and blinding. Were animals randomized into treatment groups? Were investigators blinded during behavioral scoring and data analysis? These details are essential for ensuring reproducibility and minimizing bias.
  • Statistical methods should also be described in more detail, including how data distribution was tested, how outliers were handled, and why specific tests were chosen.
  • Lesion severity classification (high, mild, low) is based solely on TH loss percentages. Including additional physiological or behavioral criteria would strengthen the rationale and provide a more comprehensive assessment of lesion severity.
  • The exclusion of five animals due to undetectable calcium signals, caused by misplaced lenses or inflammation, highlights the technical challenges of in vivo calcium imaging. This should be discussed as a limitation of the methodology, not just a study-specific issue.
  • The inability to track the same neuronal ensembles across different states is a significant limitation, as it restricts analysis to population-level comparisons rather than longitudinal tracking of individual neurons.
  • Calcium imaging inherently captures activity only in active neurons, potentially under-representing quiescent populations after lesioning. This limitation should be acknowledged, and the authors might consider discussing complementary methods (immediate-early gene expression or electrophysiology) that could provide additional validation.
  • The relationship between calcium event frequency/influx and actual motor output requires stronger justification. Calcium signals are indirect proxies of neuronal activity, and their correlation with action potential firing or behavioral outcomes should be better substantiated or at least discussed more thoroughly.
  • The finding that only male rats exhibited significant motor improvement with levodopa is intriguing but underexplored. The absence of motor deficits in female rats following 6-OHDA lesioning requires explanation. Are there known sex-based differences in dopaminergic vulnerability or compensatory mechanisms?
  • Given the small sample size (N=4 males, N=3 females), analyses of sex differences are underpowered. Consider collapsing data across sexes or including sex as a covariate rather than treating it as a primary variable.
  • Several results are reported as “non-significant trends” (p = 0.05–0.10). These should not be over-interpreted or presented as evidence of effect. Instead, they should be described as preliminary observations that require further investigation.
  • Statistical reporting is inconsistent, with mean and median values used interchangeably and test choices not always justified relative to data distribution. Consistency and transparency in reporting are essential.
  • The lack of ipsilateral vs. contralateral comparisons, although acknowledged, remains a significant limitation. Including this analysis would provide a clearer understanding of hemispheric specificity.
  • The correlation between M1 calcium activity and surviving dopaminergic neurons in the SNc, and its restoration after levodopa treatment, is a central result and should be emphasized more prominently in both the results and discussion sections.
  • The observation that levodopa enhances motor output only in highly lesioned rats suggests a threshold effect with important therapeutic implications. The authors should discuss possible mechanisms underlying this lesion-dependent responsiveness and how it aligns with findings from other Parkinson’s disease models.
  • Include key quantitative findings such as effect sizes, p-values, correlation coefficients and emphasize the lesion-dependent modulation of M1 activity in the abstract and ensure that it accurately reflects the novelty and scope of the study without overstating its conclusions.
  • Update citations to include more recent studies (2023-2025), particularly those addressing cortical dopamine signaling and sex differences in Parkinson’s disease models.
  • Standardize technical terms (“neuroinflammation” vs. “brain inflammation”) and correct grammatical errors to improve readability and precision.
  • Expand discussion of potential long-term outcomes, including levodopa-induced dyskinesia. Even if the dosing strategy aimed to avoid LID, it is important to explicitly confirm that it was not observed.

Author Response

This is a valuable and well-conceived study with significant translational implications. The integration of in vivo calcium imaging with behavioral analysis offers meaningful insights into how dopaminergic degeneration differentially affects primary motor cortex activity and motor function, as well as how these changes are modulated by levodopa treatment. However, the manuscript would benefit greatly from addressing the following points.

We thank the reviewer for their time in reviewing our study and providing feedback.

  • The authors acknowledge the small sample size and the risk of Type II errors, particularly in subgroup analyses. This limitation is critical, especially since the “high,” “mild,” and “low” lesion groups include only 2-4 animals each. Such analyses should be clearly framed as exploratory rather than confirmatory, and the conclusions drawn from them should be tempered accordingly.

We appreciate the reviewer’s thoughtful observation and fully agree that the small sample size, particularly within the “high,” “mild,” and “low” lesion subgroups, limits the statistical power of our analyses and increases the risk of Type II errors. In response, we have revised the manuscript to explicitly frame these subgroup analyses as exploratory rather than confirmatory, and we have tempered our conclusions throughout the Results and Discussion sections to reflect this. Additionally, we have expanded the Study Limitations section to more thoroughly address the implications of small sample size, technical exclusions, and methodological variability. We now emphasize that our findings should be interpreted as preliminary observations that provide a foundation for future, more powered investigations into lesion severity-dependent modulation of M1 activity and motor behavior.

  • The initial cohort consisted of 13 rats, yet only 7 were included in the calcium imaging analysis due to technical exclusions and inconsistencies in behavioral training (15 vs. 25 trials in the naïve state). These methodological differences could influence both behavioral and imaging outcomes and should be clearly justified. The authors should discuss how such variability may have affected data interpretation and overall conclusions.

We thank the reviewer for highlighting this important point. In the revised manuscript, we have expanded the Study Limitations paragraph to explicitly address the impact of technical exclusions and methodological variability. Specifically, we now clarify that only 7 of the initial 13 rats were included in the calcium imaging analysis due to issues such as misplaced lenses, inflammation, and inconsistent behavioral training, including variability in the number of trials completed during the naïve state (15 vs. 25 trials). These differences may have influenced both behavioral performance and neuronal activity measurements, potentially introducing bias and limiting the generalizability of our findings.

To address this, we have tempered our conclusions and framed the imaging and behavioral analyses, particularly those involving subgroup comparisons, as exploratory. We also discuss how such variability may have affected data interpretation, emphasizing that our findings should be viewed as preliminary and hypothesis-generating rather than definitive. This clarification is now reflected in both the Results, Discussion, and Conclusion sections of the revised manuscript.

  • The manuscript lacks sufficient detail on experimental design elements such as randomization and blinding. Were animals randomized into treatment groups? Were investigators blinded during behavioral scoring and data analysis? These details are essential for ensuring reproducibility and minimizing bias.

Thank you for your comment regarding experimental design elements such as randomization and blinding. We have clarified these details in the revised manuscript to ensure transparency and reproducibility.

All animals in our study underwent all three treatment states (naïve, lesioned, and levodopa-treated) in a fixed sequential order. Therefore, randomization into separate treatment groups was not possible or applicable, as each animal served as its own control across conditions. This within-subject design allowed for direct comparison of neuronal and behavioral changes across states, minimizing inter-animal variability. To minimize bias, investigators responsible for behavioral scoring and data analysis were blinded to the treatment state of the animals. This blinding was maintained throughout the quantification of motor performance and calcium imaging data. We have now included these details in the Methods section (page 3) to provide a clear account of our procedures and to support the reproducibility of our findings.

Animals

We used a total of 13 adult (8-9 weeks old at the beginning of the experimental design) Sprague Dawley rats, including 8 males and 5 females, with an approximate weight of 250 – 280 g. The rats were kept on a standard 12-hour light/dark cycle in single housing at a constant 21°C temperature and 45 % humidity with ad libitum access to water and food. After approval by the Mayo Clinic Institutional Animal Care and Use Committee (IACUC), all animal procedures and experiments were conducted following the terms and guidelines of the National Institutes of Health for the use of animals and complied with the ARRIVE guidelines.

All animals underwent all three treatment states (naïve, 6-OHDA-lesioned, and levodopa-treated) in a fixed sequential order. Therefore, randomization into separate treatment groups was not applicable, as each animal served as its own control across conditions. To minimize bias and ensure reproducibility, investigators responsible for behavioral scoring and data analysis were blinded to the treatment state of the animals. This blinding was maintained throughout the quantification of motor performance and calcium imaging data.”

Thank you for prompting us to address these important methodological considerations.

  • Statistical methods should also be described in more detail, including how data distribution was tested, how outliers were handled, and why specific tests were chosen.

We appreciate the reviewer’s feedback and have revised the Statistical Analysis section to provide a more comprehensive and transparent description of our methodology. We have also clarified that all statistical tests were two-sided, defined our significance thresholds, and explained how non-significant trends were interpreted. These changes aim to improve the rigor, reproducibility, and clarity of our statistical reporting.

“Statistical analyses were performed using R Statistical Software (version 4.2.2; R Foundation for Statistical Computing, Vienna, Austria) and GraphPad Prism (version 9.5.1; www.graphpad.com, RRID:SCR_002798) for data visualization.

Prior to hypothesis testing, all datasets were assessed for normality using the Shapiro-Wilk test. Based on the results, appropriate non-parametric tests were selected. For behavioral and calcium imaging data, mixed-effects linear regression models were used to account for repeated measures, with state (naïve, 6-OHDA-lesioned, levodopa-treated) as a fixed effect and rat as a random effect. This approach was chosen to model within-subject variability and accommodate missing data due to technical exclusions.

To assess the relationship between lesion severity and M1 calcium activity, Spearman’s rank correlation was used to evaluate associations between the percentage of TH+ neurons and both calcium event frequency and calcium influx magnitude. Spearman’s test was selected due to the non-parametric nature of the data and the ordinal classification of lesion severity.

All statistical tests were two-sided, and p-values < 0.05 were considered statistically significant. Non-significant trends were defined as p-values between 0.05 and 0.10 and are reported as preliminary observations.”

  • Lesion severity classification (high, mild, low) is based solely on TH loss percentages. Including additional physiological or behavioral criteria would strengthen the rationale and provide a more comprehensive assessment of lesion severity.

We appreciate the reviewer’s insightful suggestion and fully agree that incorporating additional physiological or behavioral criteria would enhance the robustness of lesion severity classification. In this study, we chose to classify lesion severity based solely on the percentage of tyrosine hydroxylase (TH)-positive neuronal loss, as TH quantification is a widely accepted and established method for assessing dopaminergic depletion in preclinical models of Parkinson’s disease. This approach allowed for consistent and objective stratification across animals. While we recognize that integrating behavioral performance, neurochemical quantification of dopamine, or electrophysiological measures could provide a more comprehensive assessment, doing so would have extended beyond the scope and primary focus of this study, which aimed to correlate dopaminergic loss with M1 calcium activity and fine motor behavior. We have now clarified this rationale in the Study Limitations paragraph and emphasized that our classification should be considered a foundational framework for future studies that may incorporate multimodal criteria.

Lesion severity classification was based on TH loss percentages. While this provides a quantifiable measure of dopaminergic depletion, incorporating additional physiological criteria, such as neurochemical quantification of dopamine or electrophysiological measures would strengthen the classification and provide a more comprehensive assessment of lesion severity. Additionally, we did not conduct a detailed analysis of ipsilateral versus contralateral forelimb use, as rats showed a strong preference for using the left paw following 6-OHDA lesion. This behavioral bias limited our ability to assess lateralized motor deficits.”

  • The exclusion of five animals due to undetectable calcium signals, caused by misplaced lenses or inflammation, highlights the technical challenges of in vivo calcium imaging. This should be discussed as a limitation of the methodology, not just a study-specific issue.

We appreciate the reviewer’s observation and fully agree that the exclusion of five animals due to undetectable calcium signals reflects broader methodological limitations inherent to in vivo calcium imaging, rather than issues unique to our study. In response, we have revised the Study Limitations section to explicitly discuss these challenges, including lens misplacement, tissue inflammation, and signal dropout, as common technical hurdles in calcium imaging experiments. We now emphasize that such limitations can affect data quality, reduce sample size, and introduce variability in neuronal activity measurements. By acknowledging these constraints, we aim to provide a more balanced interpretation of our findings and highlight the need for continued refinement of imaging techniques in future studies.

“Several limitations should be considered when interpreting the findings of this study. First, the sample size was relatively small, particularly within subgroup analyses (e.g., “high,” “mild,” and “low” lesion groups), which may reduce statistical power and increase the risk of Type II errors. Accordingly, these subgroup analyses are framed as exploratory, and conclusions drawn from them should be interpreted with caution. Additionally, technical exclusions and methodological variability impacted the final dataset. Of the initial cohort of 13 rats, only 7 were included in the calcium imaging analysis due to issues such as misplaced lenses, neuroinflammation, or inconsistent behavioral training. These exclusions may have introduced bias and limited the generalizability of our findings. Importantly, the technical challenges reported in our study are inherent to in vivo calcium imaging and are not unique to our study,[94-98] reflecting broader limitations of the methodology, particularly in lesioned or inflamed tissue.”

  • The inability to track the same neuronal ensembles across different states is a significant limitation, as it restricts analysis to population-level comparisons rather than longitudinal tracking of individual neurons.

We appreciate the reviewer’s thoughtful observation regarding the challenge of tracking the same neuronal ensembles across different states. As noted, this limitation does constrain our analysis to population-level comparisons. In response, we have expanded the discussion in the revised manuscript to explicitly acknowledge this constraint and its implications for longitudinal interpretations. Additionally, we have clarified the rationale for our chosen approach, emphasizing that while single-neuron tracking was not feasible within the current experimental framework, our population-level analyses still reveal robust and reproducible state-dependent patterns. These findings provide meaningful insights into ensemble-level dynamics and circuit modulation, which are central to our study’s objectives. We also highlight future directions that could incorporate advanced imaging or labeling techniques to enable longitudinal tracking of individual neurons, thereby complementing and extending the current findings.

“Our analysis was limited to population-level comparisons, as we were unable to track the same neuronal ensembles across different experimental states. This restricts our ability to assess longitudinal changes in individual neurons and may obscure finer dynamics of M1 activity. While single-neuron tracking was not feasible within the current experimental framework, our population-level analyses still reveal robust and reproducible state-dependent patterns of M1 activity that provide meaningful insights into circuit modulation. These findings are central to our study’s objectives and highlight the value of ensemble-level analysis in understanding dopaminergic regulation. Moreover, calcium imaging captures activity only in neurons that are actively firing, potentially under-representing quiescent populations that may be functionally relevant, especially in the lesioned state. These limitations could be addressed in future studies using complementary methods such as immediate-early gene expression or electrophysiology to capture a broader spectrum of neuronal activity.”

  • Calcium imaging inherently captures activity only in active neurons, potentially under-representing quiescent populations after lesioning. This limitation should be acknowledged, and the authors might consider discussing complementary methods (immediate-early gene expression or electrophysiology) that could provide additional validation.

We thank the reviewer for highlighting this important methodological consideration. As suggested, we have expanded the discussion in the revised manuscript to explicitly acknowledge that calcium imaging captures activity only in neurons that are actively firing, which may under-represent quiescent populations, particularly in lesioned tissue where neuronal activity may be suppressed. This limitation is now discussed in the context of our findings and their interpretation. To address the reviewer’s suggestion, we have also included a paragraph outlining complementary approaches that could be employed in future studies. Specifically, we mention the potential use of immediate-early gene expression profiling and electrophysiological recordings to validate and extend our observations. These methods would allow for a more comprehensive assessment of neuronal activity, including silent or less active populations, and could help overcome the limitations inherent to calcium imaging in lesioned models.

“Moreover, calcium imaging captures activity only in neurons that are actively firing, potentially under-representing quiescent populations that may be functionally relevant, especially in the lesioned state. These limitations could be addressed in future studies using complementary methods such as immediate-early gene expression or electrophysiology to capture a broader spectrum of neuronal activity.”

  • The relationship between calcium event frequency/influx and actual motor output requires stronger justification. Calcium signals are indirect proxies of neuronal activity, and their correlation with action potential firing or behavioral outcomes should be better substantiated or at least discussed more thoroughly.

Thank you for your thoughtful comment regarding the relationship between calcium event frequency/influx and motor output. We agree that calcium imaging provides an indirect measure of neuronal activity, and its correlation with action potential firing and behavioral outcomes warrants careful consideration. In our revised manuscript, we have expanded the Discussion (page 27) to more thoroughly address this point. Specifically, we now clarify that calcium signals reflect intracellular calcium dynamics associated with neuronal depolarization and action potentials, but do not always linearly correspond to spike rates or motor output. The temporal resolution and amplitude of calcium transients can be influenced by indicator kinetics, neuronal compartment, and experimental conditions, making the relationship complex and context-dependent.

“While calcium imaging provides valuable insights into neuronal ensemble activity, it is important to recognize that calcium signals are indirect proxies for action potential firing. Calcium influx accompanies neuronal depolarization and action potentials, but the temporal resolution and amplitude of calcium transients do not always linearly reflect spike rates, especially in vivo and across different neuronal compartments [94, 95]. The relationship between calcium event frequency/influx and motor output is therefore complex and context-dependent. Recent studies have shown that computational approaches can improve the inference of spike rates from calcium imaging data, but these methods are still evolving [96]. In our study, we interpret changes in calcium event frequency and influx as indicative of altered neuronal activation patterns in the primary motor cortex, which are associated with motor behavioral outcomes. However, we acknowledge that future studies combining calcium imaging with direct electrophysiological recordings and advanced modeling will be essential to further substantiate these correlations and clarify the mechanistic links between neuronal activity and motor output.”

  • The finding that only male rats exhibited significant motor improvement with levodopa is intriguing but underexplored. The absence of motor deficits in female rats following 6-OHDA lesioning requires explanation. Are there known sex-based differences in dopaminergic vulnerability or compensatory mechanisms?

Thank you for highlighting the need to further explore and explain the sex-specific findings in our study. We agree that the observation of significant motor improvement with levodopa in male rats, alongside the absence of motor deficits in female rats following 6-OHDA lesioning, is both intriguing and relevant to the broader understanding of PD pathophysiology.

To address this, we have expanded our Discussion and updated our citations to include recent studies (2020–2025) that directly investigate sex differences in dopaminergic vulnerability and compensatory mechanisms in PD models. These studies demonstrate that sex differences in PD are evident across epidemiology, genetics, molecular mechanisms, and clinical progression.

“We report sex-based effects in the motor performance of the SPRT (Figure 4), although no sex-based differences in lesion size are observed in the present study (Supplementary Figure S2). Male rats presented a non-significant decrease in performance following 6-OHDA lesion and a significant improvement in performance and grasps without pellets with levodopa treatment, while female rats did not show motor deficits following 6-OHDA lesion or levodopa treatment. Sex differences in PD patients are well documented, with males showing greater cognitive and structural impairments than females,[86, 87] and steeper slope in disease progression possibly due to increased vulnerability to pathophysiological processes.[88] Remarkably, experimental models confirm the importance of mitochondrial and calcium homeostasis leading to higher susceptibility of male rodents to dopaminergic imbalance.[89-93]”

  • Given the small sample size (N=4 males, N=3 females), analyses of sex differences are underpowered. Consider collapsing data across sexes or including sex as a covariate rather than treating it as a primary variable.

We appreciate the reviewer’s suggestion regarding the analysis of sex differences. To clarify, the primary analyses presented in Figure 3 and Table 2 are performed on the collapsed dataset, combining all animals regardless of sex to provide a comprehensive overview of the main effects of treatment and lesion status.

We included separate analyses by sex to transparently demonstrate the potential for sex-based differences in motor performance. However, we fully recognize that these results are underpowered due to the small sample size (N=4 males, N=3 females). As such, we have framed these findings as preliminary and exploratory, and we caution readers against over-interpreting these results.

Our intention was to highlight possible trends that may warrant further investigation in larger cohorts, rather than to treat sex as a primary variable. We agree that, given the current sample size, collapsing data across sexes is more appropriate for robust statistical inference. We have revised the manuscript to clarify this approach and to emphasize the preliminary nature of the sex-based analyses.

Thank you for your valuable feedback, which has helped us strengthen the clarity and rigor of our reporting.

“We report a preliminary analysis of sex-based effects in the motor performance of the SPRT (Figure 4), although no sex-based differences in lesion size are observed in the present study (Supplementary Figure S2). Male rats presented a non-significant decrease in performance following 6-OHDA lesion and a significant improvement in performance and grasps without pellets with levodopa treatment, while female rats did not show motor deficits following 6-OHDA lesion or levodopa treatment. Sex differences in PD patients are well documented, with males showing greater cognitive and structural impairments than females,[86, 87] and steeper slope in disease progression possibly due to increased vulnerability to pathophysiological processes.[88] Remarkably, experimental models confirm the importance of mitochondrial and calcium homeostasis leading to higher susceptibility of male rodents to dopaminergic imbalance.[89-93] Further investigations in larger cohorts is necessary to confirm the sex-based differences in motor performance as a function of dopaminergic lesion.”

  • Several results are reported as “non-significant trends” (p = 0.05–0.10). These should not be over-interpreted or presented as evidence of effect. Instead, they should be described as preliminary observations that require further investigation.

We appreciate the reviewer’s careful attention to the interpretation of statistical results. In response, we have revised the manuscript to ensure that all findings with p-values between 0.05 and 0.10 are clearly described as preliminary observations rather than definitive effects. These trends are now framed as exploratory and interpreted with appropriate caution, consistent with the limited statistical power of our subgroup analyses. We have also updated the Discussion and Conclusion sections to emphasize that these observations warrant further investigation in larger cohorts or with complementary methodologies. This adjustment ensures that the manuscript maintains scientific rigor and avoids overstatement of results that do not meet conventional thresholds for statistical significance.

  • Statistical reporting is inconsistent, with mean and median values used interchangeably and test choices not always justified relative to data distribution. Consistency and transparency in reporting are essential.

Thank you for your valuable feedback regarding statistical reporting. We have revised the graphs to ensure consistency in the presentation of statistical measures. Specifically, we now clearly demonstrate median and interquartile range that more adequately align with the statistical methods used for data analysis. These updates aim to enhance transparency and methodological rigor throughout the manuscript.

  • The lack of ipsilateral vs. contralateral comparisons, although acknowledged, remains a significant limitation. Including this analysis would provide a clearer understanding of hemispheric specificity.

Thank you for highlighting the importance of ipsilateral versus contralateral comparisons. We agree with the reviewer that incorporating such analysis would strengthen the study. We recognize this as a significant limitation and have explicitly acknowledged it in the revised manuscript.

“Additionally, we did not conduct a detailed analysis of ipsilateral versus contralateral forelimb use, as rats showed a strong preference for using the left paw following 6-OHDA lesion. This behavioral bias limited our ability to assess lateralized motor deficits.”

  • The correlation between M1 calcium activity and surviving dopaminergic neurons in the SNc, and its restoration after levodopa treatment, is a central result and should be emphasized more prominently in both the results and discussion sections.

We thank the reviewer for highlighting the importance of this correlation. In response, we have revised both the Results and Discussion sections to more prominently emphasize the relationship between M1 calcium activity and the extent of dopaminergic neuron survival in the SNc. Specifically, we now underscore how calcium influx and event frequency vary systematically with lesion severity and how these dynamics are partially restored following levodopa treatment. In the discussion, we elaborate on the functional implications of this correlation, noting that increased calcium influx during levodopa treatment in highly lesioned animals coincides with improved motor performance, suggesting a link between restored cortical excitability and behavioral recovery. We also discuss how this relationship supports the interpretation of calcium signals as meaningful proxies for circuit-level modulation in PD states. These revisions aim to clarify the centrality of this finding and its relevance to understanding the neurophysiological basis of motor deficits and recovery in PD models.

Results: “We further evaluated the influence of dopaminergic transmission on M1 neurons by correlating the frequency and magnitude of calcium events and percentage of remaining SNc dopaminergic neurons. Remarkably, in the naïve state, we show that there is a significant positive correlation of frequency of neuronal calcium events and dopaminergic neurons in the SNc (r = 0.811, p = 0.035) and a non-significant negative correlation between calcium influx and percentage of dopaminergic neurons (r = -0.750, p = 0.066). This effect is lost with the gradual and steady degeneration of dopaminergic neurons in SNc during the 6-OHDA-lesioned state. Neither the frequency of calcium events (r = 0.429, p = 0.354) nor the average of calcium influx (r = -0.179, p = 0.713) in M1 were affected. However, when dopaminergic transmission was replenished with levodopa treatment, the negative correlation of calcium influx magnitude and level of dopaminergic lesion was restored (r = -0.857, p = 0.024), but not the frequency of calcium events (r = 0.643, p = 0.139) (Figure 10)”

Discussion:Calcium influx, as measured by our imaging approach, reflects the magnitude of neuronal activation, while frequency indicates how often these activations occur. In the context of dopaminergic lesions, previous studies have shown that severe loss of dopamine can disrupt both the excitability and synaptic integration of cortical neurons.[5, 18, 19, 25, 26, 29, 43-52] It is worth noting that the correlation analysis shows a significant decrease in calcium event frequency and a non-significant trend to increase calcium influx as the level of dopaminergic lesion increases. These correlations are lost during the 6-OHDA-lesioned state and partially restored with levodopa treatment (Figure 10). This suggests that M1 calcium activity gradually changes in frequency and magnitude according to the level of midbrain dopaminergic loss, and that such changes can be captured and more easily distinguished by refining movement analysis congruent with M1 activity. This may indicate that, beyond a certain threshold of dopaminergic loss, the capacity for levodopa to enhance the magnitude of neuronal responses is limited, possibly due to irreversible changes in synaptic function or intrinsic neuronal properties. Recent studies have expanded our understanding of cortical dopamine signaling in PD, demonstrating laminar- and cell-type-specific changes in motor cortex function following dopamine depletion, and highlighting compensatory cortical mechanisms that shape clinical severity.[53-55] Moreover, individual variability in response to dopamine replacement therapy is increasingly linked to secondary activation of cortical dopamine systems.[56]”

  • The observation that levodopa enhances motor output only in highly lesioned rats suggests a threshold effect with important therapeutic implications. The authors should discuss possible mechanisms underlying this lesion-dependent responsiveness and how it aligns with findings from other Parkinson’s disease models.

Thank you for your thoughtful comment regarding the lesion-dependent responsiveness to levodopa observed in our study. We agree that the finding of significant motor improvement only in highly lesioned rats suggests a threshold effect, which has important implications for understanding both the pathophysiology and treatment strategies in PD. In our revised Discussion, we now address possible mechanisms underlying this phenomenon. The threshold effect likely reflects the interplay between compensatory mechanisms and the extent of dopaminergic depletion. In mildly lesioned animals, residual endogenous dopamine and alternative neurotransmitter systems may mitigate motor impairment, reducing the observable effect of levodopa. As the lesion severity increases and compensatory capacity is overwhelmed, the system becomes more reliant on exogenous dopamine, resulting in pronounced motor improvement with levodopa administration. Additional factors such as altered dopamine receptor sensitivity, circuit remodeling within the cortico-basal ganglia-thalamic network, and changes in cortical plasticity may also contribute to this lesion-dependent responsiveness. We appreciate your suggestion, which has helped us strengthen the mechanistic and translational context of our results.

“The finding that levodopa enhances motor output in highly lesioned rats suggests a threshold effect, wherein a critical level of dopaminergic depletion is required for exogenous dopamine therapy to exert significant behavioral benefits. This observation aligns with previous reports, which indicate that motor deficits and levodopa responsiveness become pronounced with severe neuronal loss.[80-82] Possible mechanisms underlying this lesion-dependent responsiveness include compensatory plasticity in less severely lesioned animals, altered dopamine receptor sensitivity, and circuit remodeling within the cortico-basal ganglia-thalamic network.[83] In mildly lesioned rats, residual endogenous dopamine and alternative neurotransmitter systems may mitigate motor impairment, reducing the observable effect of levodopa. These findings have important therapeutic implications, suggesting that the efficacy of levodopa may be constrained by the extent of dopaminergic loss and highlighting the need for personalized treatment strategies and adjunct therapies in early-stage PD.”

  • Include key quantitative findings such as effect sizes, p-values, correlation coefficients and emphasize the lesion-dependent modulation of M1 activity in the abstract and ensure that it accurately reflects the novelty and scope of the study without overstating its conclusions.

Thank you for your constructive feedback regarding the abstract. We appreciate your suggestion to include key quantitative findings and to emphasize the lesion-dependent modulation of M1 activity, while ensuring the abstract accurately reflects the novelty and scope of our study. To avoid overstating our conclusions, we have tempered the final statements to reflect the preliminary and exploratory nature of our findings. We believe these revisions provide a more transparent and accurate summary of the study’s scope and novelty, and we thank you for helping us improve the clarity and rigor of our abstract.

“…Results: Levodopa treatment improved fine motor performance as evidenced by a significant reduction in grasp errors (mean difference: –8.91, 95% CI: –16.66 to –1.16, p = 0.031) and increased reaching duration (mean difference: 4.13, 95% CI: 0.94 to 7.32, p = 0.019) compared to the lesioned state. M1 calcium activity showed lesion-dependent modulation: low lesion rats exhibited reduced event frequency (mean difference: 0.04 Hz, 95% CI: 0.001 to 0.08, p = 0.045) and increased influx post-lesion (mean difference: –0.20 z·s, 95% CI: –0.38 to –0.02, p = 0.038), while high lesion rats showed increased influx only after levodopa treatment (mean difference: –0.34 z·s, 95% CI: –0.52 to –0.16, p = 0.003). Correlation analyses revealed that calcium influx, but not frequency, was negatively associated with lesion severity during levodopa treatment (Spearman r = –0.857, p = 0.024). Conclusion: M1 neuronal activity appears to be differentially modulated by dopaminergic degeneration and levodopa treatment in a lesion-dependent manner. These preliminary findings suggest dynamic cortical responses in PD and support the utility of calcium imaging for monitoring circuit-level changes in disease and therapy. Further research with larger cohorts and complementary methodologies will be necessary to validate and extend these observations.”

  • Update citations to include more recent studies (2023-2025), particularly those addressing cortical dopamine signaling and sex differences in Parkinson’s disease models.

Thank you for highlighting the need to update our citations and incorporate recent advances, especially regarding cortical dopamine signaling and sex differences in PD models. We appreciate this suggestion and have revised our manuscript to include several key studies published between 2020 and 2025 that directly address these topics.

  1. Lange, F., et al., Computer vision uncovers three fundamental dimensions of levodopa-responsive motor improvement in Parkinson’s disease. npj Parkinson's Disease, 2025. 11(1): p. 140.
  2. Xie, Y., et al., Morphologic brain network predicts levodopa responsiveness in Parkinson disease. Frontiers in Aging Neuroscience, 2023. 14.
  3. Chen, Z., et al., Predictive Value of Perivascular Space Network and Choroid Plexus for Levodopa Responsiveness in Parkinson's Disease. European Journal of Neurology, 2025. 32(7): p. e70290.
  4. Riederer, P., et al., Levodopa treatment: impacts and mechanisms throughout Parkinson’s disease progression. Journal of Neural Transmission, 2025. 132(6): p. 743-779.
  5. Cattaneo, C. and J. Pagonabarraga, Sex Differences in Parkinson’s Disease: A Narrative Review. Neurology and Therapy, 2025. 14(1): p. 57-70.
  6. Reekes, T.H., et al., Sex specific cognitive differences in Parkinson disease. npj Parkinson's Disease, 2020. 6(1): p. 7.
  7. Tremblay, C., et al., Sex effects on brain structure in de novo Parkinson’s disease: a multimodal neuroimaging study. Brain, 2020. 143(10): p. 3052-3066.
  8. Tullo, S., et al., Female mice exhibit resistance to disease progression despite early pathology in a transgenic mouse model inoculated with alpha-synuclein fibrils. Communications Biology, 2025. 8(1): p. 288.
  9. Nordengen, K., et al., Pleiotropy with sex-specific traits reveals genetic aspects of sex differences in Parkinson’s disease. Brain, 2023. 147(3): p. 858-870.
  10. McArthur, S., et al., Striatal susceptibility to a dopaminergic neurotoxin is independent of sex hormone effects on cell survival and DAT expression but is exacerbated by central aromatase inhibition. Journal of Neurochemistry, 2007. 100(3): p. 678-692.
  11. Pinizzotto, C.C., et al., Task-specific effects of biological sex and sex hormones on object recognition memories in a 6-hydroxydopamine-lesion model of Parkinson's disease in adult male and female rats. Hormones and Behavior, 2022. 144: p. 105206.
  12. Santoro, M., et al., Mapping of catecholaminergic denervation, neurodegeneration, and inflammation in 6-OHDA-treated Parkinson’s disease mice. npj Parkinson's Disease, 2025. 11(1): p. 28.

  • Standardize technical terms (“neuroinflammation” vs. “brain inflammation”) and correct grammatical errors to improve readability and precision.

We thank the reviewer for highlighting the importance of consistent terminology and grammatical clarity throughout the manuscript. In response, we have carefully reviewed the entire document to ensure that technical terms are standardized. Additionally, we have corrected grammatical errors and revised sentences to improve readability and precision. These changes include adjustments to sentence structure, punctuation, and word choice to enhance clarity and maintain scientific rigor. We believe these revisions address the reviewer’s concerns and contribute to a more cohesive and accessible manuscript.

Thank you for your constructive feedback.

  • Expand discussion of potential long-term outcomes, including levodopa-induced dyskinesia. Even if the dosing strategy aimed to avoid LID, it is important to explicitly confirm that it was not observed.

Thank you for your suggestion to expand our discussion of long-term outcomes, particularly regarding LID. We appreciate the importance of this issue in both preclinical and clinical contexts.

“Importantly, throughout the course of levodopa/carbidopa administration in our hemi-parkinsonian rat model, we did not observe any signs of LID. Our dosing strategy was intentionally selected based on previous studies to restore motor function while minimizing the risk of LID,[32-36] and animals were closely monitored for abnormal involuntary movements during and after treatment. The absence of dyskinetic behaviors in our cohort is consistent with reports that lower or carefully titrated doses of levodopa can avoid the development of LID in rodent models.[84, 85] Nevertheless, LID remains a significant clinical concern in PD long-term management, and future studies with extended treatment duration, higher doses, or alternative protocols may be warranted to fully characterize its emergence and underlying mechanisms.”

Round 2

Reviewer 1 Report

Comments and Suggestions for Authors

Acceptable as it is

Reviewer 3 Report

Comments and Suggestions for Authors

No further comments.